# Harbouring public good mutants within a pathogen population can increase both fitness and virulence

Richard J Lindsay, Michael J Kershaw, Bogna J Pawlowska, Nicholas J Talbot, Ivana Gudelj*

School of Biosciences, University of Exeter, Exeter, United Kingdom

**Abstract** Existing theory, empirical, clinical and field research all predict that reducing the virulence of individuals within a pathogen population will reduce the overall virulence, rendering disease less severe. Here, we show that this seemingly successful disease management strategy can fail with devastating consequences for infected hosts. We deploy cooperation theory and a novel synthetic system involving the rice blast fungus *Magnaporthe oryzae*. In vivo infections of rice demonstrate that *M. oryzae* virulence is enhanced, quite paradoxically, when a public good mutant is present in a population of high-virulence pathogens. We reason that during infection, the fungus engages in multiple cooperative acts to exploit host resources. We establish a multi-trait cooperation model which suggests that the observed failure of the virulence reduction strategy is caused by the interference between different social traits. Multi-trait cooperative interactions are widespread, so we caution against the indiscriminant application of anti-virulence therapy as a disease-management strategy.

*For correspondence: I.Gudelj@ exeter.ac.uk

**Competing interests:** The authors declare that no competing interests exist.

## Introduction

Targeting virulence to disarm rather than to eradicate pathogens, is a nascent disease management strategy that has been proposed to slow the evolution of antibiotic resistance. Virulence reduction strategies deploy drugs (*Clatworthy et al., 2007*; *Rasko and Sperandio, 2010*) but despite some success in clinical trials (*Lowy et al., 2010*), drug-resistance has already been observed (*García-Contreras et al., 2013*). An alternative strategy is to use live organisms to reduce pathogen virulence through competitive displacement. This idea is particularly promising; proving successful in some animal (*Harrison et al., 2006*; *Rumbaugh et al., 2009*; *Pollitt et al., 2014*) and plant (*Frey et al., 1994*) models, as well as clinical (*Gerding et al., 2015*) and agricultural field studies (*Amaike and Keller, 2011*; *Cotty, 1990*; *Cotty and Bayman, 1993*).

During competitive displacement treatments, strains with attenuated virulence are introduced into hosts, be they patients or crops, to out-compete highly virulent pathogens. Indeed, recent clinical trials exploited a low virulence, non-toxigenic strain of *Clostridium difficile* in patients to out-compete virulent toxin-producing strains (*Gerding et al., 2015*). Similar treatments are commercially available in agriculture (*Amaike and Keller, 2011*) where a low virulence non-toxigenic strain of *Aspergillus flavus* is used to prevent virulent toxin-producing strains from infecting crops through competitive exclusion (*Cotty, 1990*; *Cotty and Bayman, 1993*). Even cancers are known to evolve heterogeneous cell populations and, as a result, the exploitation of competitive interactions between tumour cells is seen as a promising disease treatment (*Korolev et al., 2014*; *Jansen et al., 2015*; *Tabassum and Polyak, 2015*). Is the competitive displacement strategy failsafe, however, or is there a risk that such interventions might have unforeseen consequences? Here, we address this question using cooperation theory and a new synthetic system involving the rice blast fungus.

Successful infection and proliferation of a pathogen is frequently aided by cooperation between individual microbial cells. The most common type of pathogen cooperation involves production of extracellular factors used to perform a range of functions that directly or indirectly contribute to pathogen virulence and so are referred to as virulence factors. Direct virulence factors aid pathogenesis by directly interacting with and damaging the host, such as by the production of infection structures or toxins (*Raymond et al., 2012*; *West et al., 2007*). Alternatively, indirect virulence factors facilitate the survival and proliferation of the pathogen within the host. Examples include products for nutrient acquisition, suppression or evasion of host immunity, antibiotic resistance, biofilm formation and behaviours which can be coordinated by quorum sensing molecules (*Harrison et al., 2006*; *Rumbaugh et al., 2009*; *Pollitt et al., 2014*; *West et al., 2007*; *Köhler et al., 2009a*; *Lee et al., 2010*). Such extracellular factors can be considered public goods, because they benefit every individual in the locality and are therefore open to exploitation by cheats who do not contribute to the cost of their production, but still reap the rewards. Since public goods aid microbial growth and survival, they also affect the extent of damage that pathogens can inflict on their hosts. As a consequence, cooperators that produce extracellular factors are often more virulent than non-producing mutants (*Harrison et al., 2006*; *Rumbaugh et al., 2009*; *Pollitt et al., 2014*; *Raymond et al., 2012*; *Köhler et al., 2009a*; *Buckling and Brockhurst, 2008*).

To develop our synthetic rice blast fungus system we exploit two key ingredients of cooperation theory: the ability of cheats to invade populations of cooperators by not paying the cost of cooperation, but reaping the benefits, and the fact that public good non-producing mutants are often less virulent than public good-producing co-operators. In that case, cooperation theory predicts that the presence of public good cheats with low virulence within a population of virulent public good cooperators leads to a reduction in overall virulence of the population (*Buckling and Brockhurst, 2008*). It is important to note that this does not apply to systems where cheats are more virulent than cooperators, as in the case of prudent resource use, since an introduction of cheats with high virulence into a population of cooperators with low virulence will lead to an increase in population virulence (*Bremermann, 1983*; *Nowak and May, 1994*; *Frank, 1996*).

Rice blast is the most destructive disease of cultivated rice (*Oryza sativa*) and is caused by the filamentous fungus *Magnaporthe oryzae*, which is also a leading model system for studying host-parasite interactions (*Wilson and Talbot, 2009*). *M. oryzae* infections are polycyclic, initiated by infectious agents (spores called conidia) that exploit the host plant to replicate and form new agents for transmission (*Wilson and Talbot, 2009*). To identify a cooperative public good-producing trait, we focused on traits that enable the fungus to exploit nutrients inside host tissue for proliferation and spore production, which is widely used as a proxy for pathogen fitness (see Materials and methods). Given that the most abundant storage sugar within plant tissue is sucrose, we reasoned that secreted invertase production might represent a social trait and an indirect virulence factor. Invertase catalyses the hydrolysis of sucrose into glucose and fructose, which are preferred carbon sources, that are then transported into *M. oryzae* by hexose transporters for metabolism (*Talbot, 2010*). We reasoned that in a mixed population, an invertase mutant may behave as a 'cheat' because it can exploit the monosaccharides liberated by invertase-secreting individuals. Here we test the competitive displacement anti-virulence strategy by investigating how fungal virulence is affected in mixed population of cooperators and cheats exhibiting different levels of virulence.

## Results

First, we set out to determine whether invertase production is a cooperative trait in *M. oryzae*. To do this we generated a mutant, Δ*inv1*, by targeted deletion of the *INV1* gene (MGG_05785), which encodes the major invertase activity in the wild type strain of the fungus, Guy11, (*Figure 1a–c*, *Figure 1—figure supplements 1* and *2*). This is, to our knowledge, the first invertase mutant generated in *M. oryzae*. However, the gene deletion of *INV1* that we created synthetically could potentially occur in nature because mutation, gene deletion and transposon-mediated gene disruption are all frequently found in wild populations of the fungus (*Farman et al., 2002*; *Kang et al., 2001*; *Bonman, 1992*). Moreover, natural fungal populations have been reported to show a high degree of intra-specific diversity in invertase activity (*Naumov et al., 1996*; *El-Said, 2002*), suggesting that variation in this trait occurs in the wild.

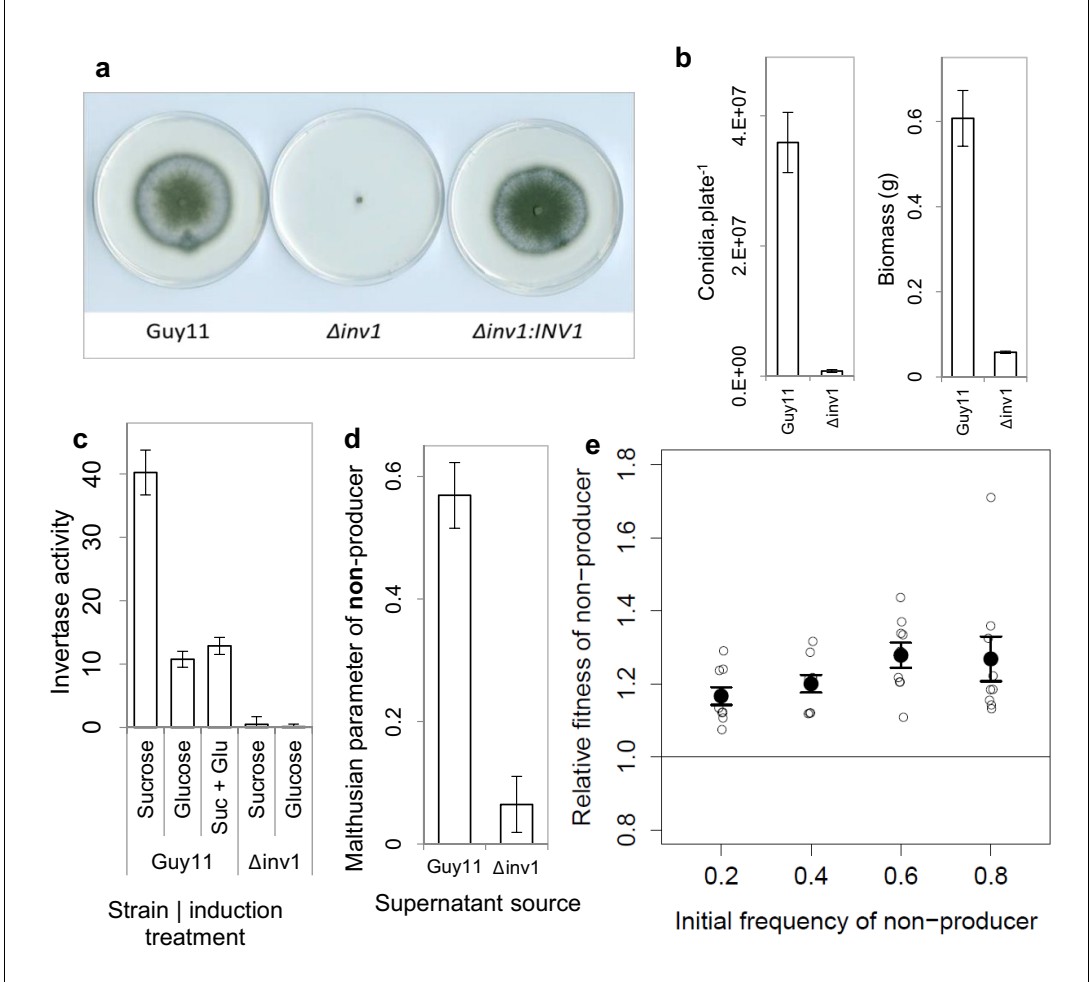

**Figure 1.** Invertase production in *M. oryzae* is a cooperative trait. (a) Δ*inv1* has growth defects on sucrose minimal media, with functional complementation restoring invertase synthesis and growth morphology of Δ*inv1:INV1*, confirming the function of *INV1*. (b) Invertase deficiency resulted in a fitness reduction on sucrose (mean ± s.e.m.) with respect to conidia (p<0.0001, n = 12, two-sided 2-sample t-test for unequal variance) and biomass (p<0.0001, n = 9, two-sided 2-sample t-test for unequal variance). (c) This was confirmed to be caused by invertase production deficiency tested by enzymatic assay of culture filtrate under different induction treatments (units are μ moles of glucose / fructose liberated from sucrose per minute) mean ± s.e.m., n = 3. *INV1* expression in Guy11 is sucrose induced and glucose repressed, with constitutive expression remaining in non-yielding environments (glucose). (d) *INV1* production is an exploitable secreted product as Δ*inv1* could generate significantly more biomass in the supernatant of Guy11 than in the supernatant of Δ*inv1* (p<0.0001, n = 9, two-sample t-test for equal variance). (e) The non-producer, Δ*inv1*, gains a fitness advantage over invertase producers in a low-structured environment (p<0.003 at each initial frequency, one-sample t-test, n = 9, mean ± s.e.m.). A small amount of x-axis noise was added to help visualize data points.

The following figure supplements are available for figure 1:

**Figure supplement 1.** Targeted gene deletion of *INV1* in *Magnaporthe oryzae*.

**Figure supplement 2.** Functional complementation of *Saccharomyces cerevisiae* invertase deletion strain DBY1701 with *M. oryzae INV1*.

**Figure supplement 3.** The relative fitness of producers (Guy11) in a spatially structured population.

The Δ*inv1* mutant showed impaired ability to grow on sucrose, as measured by spore production and biomass formation (*Figure 1a,b*), resulting from reduced secreted invertase activity (*Figure 1c*). This confirmed the function of *INV1* and demonstrated the benefit of the public good to populations of producers. To establish that invertase is exploitable by non-producers, we showed that Δ*inv1* mutants can recover their ability to grow on sucrose, provided that invertase activity is provided by

the presence of an isogenic wild type strain Guy11 (*Figure 1d*). As predicted by social evolution theory (*Frank, 1998*), the non-producers also gain a fitness advantage in mixed populations, by exploiting the public good generated by the producer, while avoiding the cost of its production (*Figure 1e*). Furthermore, in sufficiently structured environments, the producer can gain a selective advantage when it is rare in a population consisting predominantly of non-producers (*Figure 1—figure supplement 3*). In this case the fitness of the producer is negative-frequency-dependent and the coexistence of both producers and non-producers is possible at intermediate frequency. This is consistent with previously studied invertase production systems (*MacLean et al., 2010*). The fitness measure used to produce frequency dependence plots assumes constant fitness differences between competitors and therefore is not necessarily a predictor of long-term equilibrium frequencies (*Ribeck and Lenski, 2015*).

Invertase production can also be considered an exploitable social trait during plant infections, with *INV1* contributing to *M. oryzae* fitness and virulence. The △*inv1* mutant showed dramatically reduced fitness during plant infections when present in isolation, as measured by conidial production at the end of an infection cycle (*Figure 2a*) and also exhibited lower virulence, measured by the area of disease lesion coverage of an infected leaf (*Figure 2b*, *Figure 2—figure supplement 1*). These lesions are symptomatic of rice blast disease (*Wilson and Talbot, 2009*) and are a direct sign of damage inflicted upon the host, affecting host growth and survival. In mixed infections, live cell imaging of diseased rice tissue showed that the two isogenic strains, which expressed green and red fluorescent protein-encoding reporter genes to allow them to be easily distinguished from one another (*Figure 2—figure supplement 2*), infected the same or neighbouring host plant cells (*Figure 2c*). This suggests that Δ*inv1* mutants are able access the public good generated by a co-infecting producer, Guy11. In addition, Δ*inv1* had a selective advantage over the producer in mixed infections (relative fitness of Δ*inv1* at 20% initial frequency was $v = 2.11 \pm 0.28$ s.e.m. n = 32), suggesting that invertase production is costly and can therefore be exploited by a non-producer. This observation also predicts that in the long-term non-producers will not be eliminated from mixed strain populations.

Our findings show that invertase production is a cooperative trait in *M. oryzae,* with the invertase producer, Guy11, termed a cooperator and the non-producer mutant, △*inv1,* termed a cheat. Moreover, Guy11 cooperator is more virulent than △*inv1* cheat.

We then asked what would happen if we inoculated rice with a mixed population of the virulence-impaired △*inv1* mutant with a fully pathogenic isogenic wild type strain, Guy11. Infection studies were conducted over nine days following an entire rotation in a polycyclic disease (*Figure 2—figure supplement 3*), using infection assays that are comparable to wild infections (see Materials and methods). According to all prior theory and in vivo infection experiments, we expected that the virulence of the population would decrease in such an infection (*Harrison et al., 2006*; *Rumbaugh et al., 2009*; *Pollitt et al., 2014*; *Köhler et al., 2009a*; *Buckling and Brockhurst, 2008*; *Brown et al., 2009*; *Crespi et al., 2014*). Strikingly, we observed the opposite result. Indeed, infecting populations composed of just the highly virulent strain were not the fittest (*Figure 2a*, *Figure 2—figure supplements 1* and *4*), compared to the mixed infections. We then tested if this increased fitness in mixed infections translated into more damage to the host. Again, infecting populations comprised exclusively of the highly virulent invertase producers were not the most virulent (*Figure 2b*, *Figure 2—figure supplement 1*). Therefore, introducing less virulent non-producers into a population of highly virulent producers can lead to an increase in the production of new spores (*Figure 2a*) and crucially to an increase in the damage to the host (*Figure 2b*), compared to infections consisting only of the highly virulent producers. Moreover, the new spores generated at the end of the mixed infection, which instigate a new infection cycle, contained both producer and non-producer strains (*Figure 2d*, *Figure 2—figure supplement 5*).

We next set out to understand why our observations contradict the current understanding behind virulence reduction strategies. We postulated that this might result from multiple-interacting social traits (*MacLean et al., 2010*; *Brown and Taylor, 2009*). The impact of public goods on virulence has only ever been considered in isolation, without considering the influence of additional social dilemmas facing the pathogen. We reasoned that alongside public goods production, *M. oryzae* faces a second social dilemma of self-restraint or 'tragedy of the commons' (*Hardin, 1968*), which requires a pathogen to convert available resources into energy slowly but efficiently, as opposed to rapidly and inefficiently. When we investigated this possibility, we found evidence of a rate-efficiency trade-off

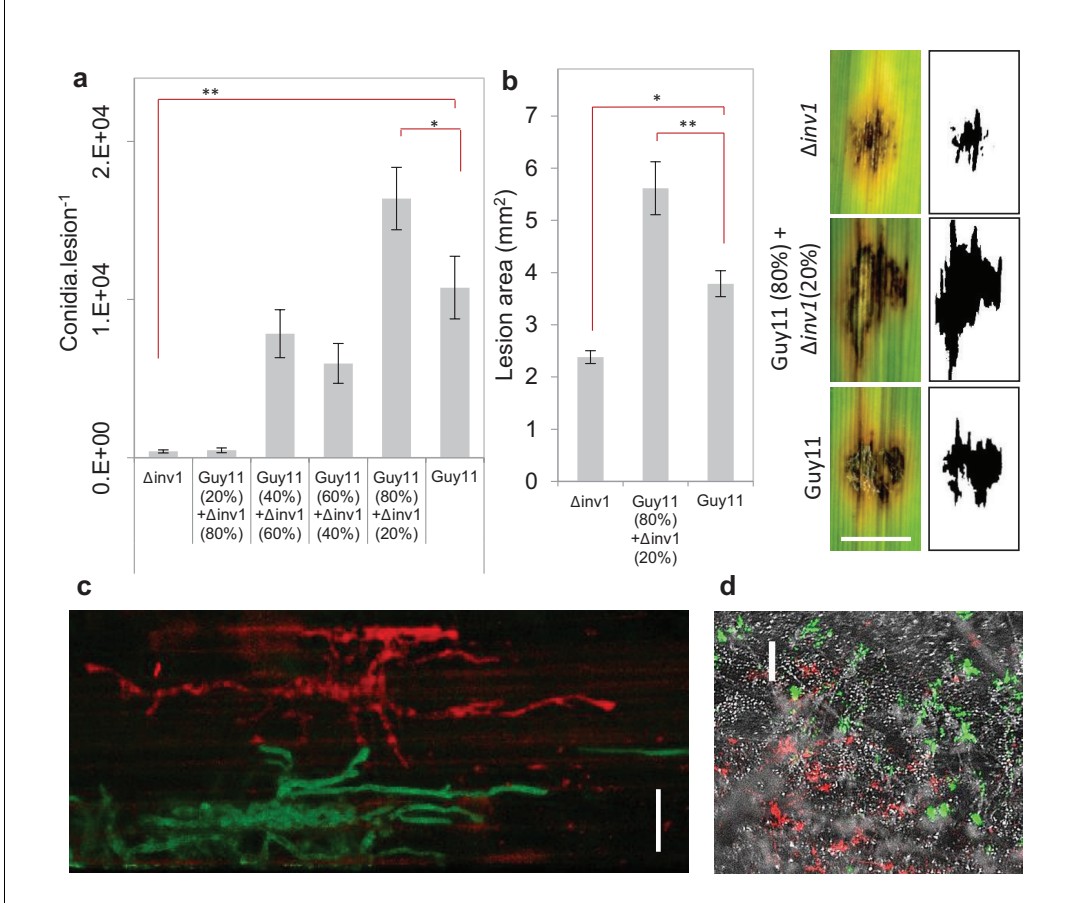

**Figure 2.** Virulence and pathogen fitness measurements of Δ*inv1*, Guy11 and a mixed inoculum. (a) *In planta* fitness of *M. oryzae* during infection was evaluated by leaf spot inoculation (mean ± s.e.m., n = 42). Fitness was quantified by the number of conidia recovered per lesion at the end of the disease cycle. Infections with Guy11 produced significantly more conidia than pure non-producer (Δ*inv1*) infections (**p<0.0001, W = 66, two-sided Mann-Whitney U test, n = 42). In addition, applying existing social theory we hypothesize that the number of conidia recovered per mixed Guy11 and Δ*inv1* disease lesions is not higher than the number of conidia recovered from Guy11-only disease lesions. However, we can reject this hypothesis using properties of Boolean algebra: analysis of raw data (*p<0.0365, W = 1174, two-sided Mann-Whitney U-test with Bonferroni correction, n = 42, see Appendix 1A for detailed analysis) and log-transformed data (*Figure 2—figure supplement 4*). (b) Disease virulence of *M. oryzae* during infection was also evaluated by spot inoculation (mean±s.e.m., n = 20). It showed reduced virulence, as measured by lesion area, of Δ*inv1* compared to Guy11 (*p<0.00003, two-sided 2-sample t-test for unequal variances, mean±s.e.m., n = 20). Guided by the data in panel (a), we also confirm that fitness positively correlates with virulence and that mixed populations of Guy11 and Δ*inv1* also have higher virulence than pure Guy11 infections (**p<0.0032, two-sided 2-sample t-test for unequal variances, n = 20). Example lesions (7d) from leaf spot infections from pure and mixed populations, scale bar = 3 mm, with ImageJ analysis of images from which lesion areas were measured. Images of all replicates can be seen in *Figure 2—figure supplement 1*. (c) Live cell imaging of mixed strain infection (48 hr.p.i.) of rice sheath epidermal cells indicating close proximity of co-infecting strains; this suggests interactions and invertase exploitation is possible, scale bar = 50 μm. (d) Epifluorescence micrograph of sporulating lesion from mixed infections (9d) with DIC, RFP (wildtype) and GFP (Δ*inv1*) conidia, indicating the presence of both strains within conidia populations at the end of the infection cycle, scale bar = 200 μm.

The following figure supplements are available for figure 2:

**Figure supplement 1.** Extended summary of *Figure 2*.

**Figure supplement 2.** Strains were distinguishable by the presence of fluorescent protein tag.

**Figure supplement 3.** The disease cycle of *Magnaporthe oryzae*.

**Figure supplement 4.** Log-transformation of the data in *Figure 2a* showing pathogen fitness measurement of Δ*inv1*, Guy11 and a mixed inoculum.

**Figure supplement 5.** Micrograph of sporulating lesion from mixed strain leaf spot infection.

(**Meyer et al., 2015**) whereby faster-growing populations were less efficient in spore production per unit of carbon resource (**Figure 3**), suggesting a tragedy of the commons scenario can occur in sugar utilisation by the pathogen during plant infection. The multi-trait interactions matter most if one considers the spatial and temporal trajectory of population growth. When producers are common, invertase production is expected to result in a large spike, both spatial and temporal, in available glucose. This would enable rapid but inefficient growth of the pathogen. However, if a fraction of non-producers is introduced into the population, the glucose spike around producers in the vicinity of non-producers would be smaller, such that the population would consume finite resources more efficiently. This would lead to fitter and more virulent populations, as observed in **Figure 2a–b**.

Is the synergy between public goods production and self-restraint sufficient to explain the enhanced fitness and virulence in a mixed infection of invertase producers and non-producers? To test this idea, we generated a mathematical model incorporating both social traits and taking into account the inherently spatial nature of plant infections (**Figure 2c–d**; for model details see Appendix 1B and **Figure 4—figure supplement 1a**). The model successfully produced the unexpected empirical result (**Figure 4—figure supplement 1b**) and also predicted that the presence of the

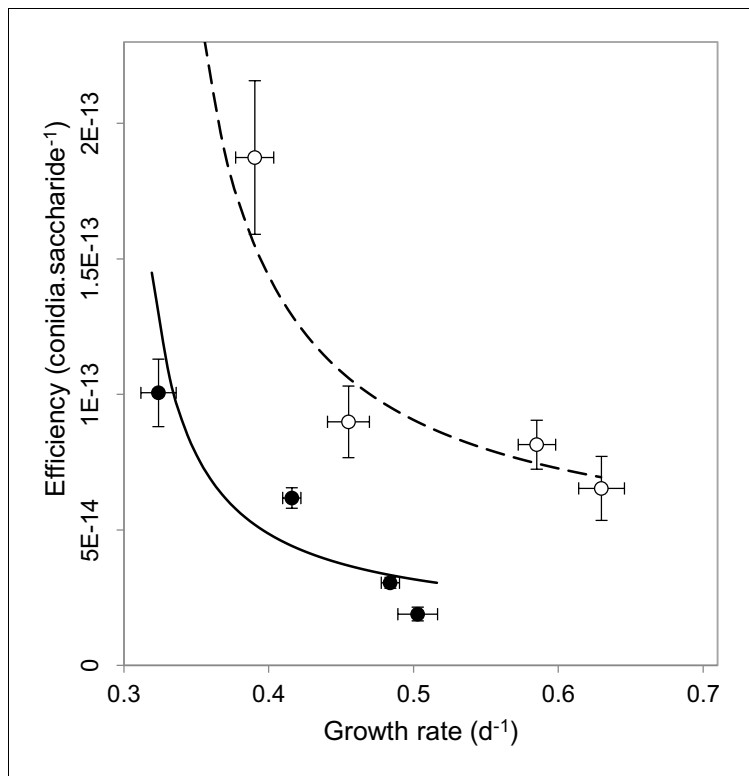

**Figure 3.** Multi-trait interactions during sucrose metabolism by *M. oryzae*. In addition to public good invertase production (**Figure 1**), we found evidence of a rate-efficiency trade-off where resources are used less efficiently when abundant, applicable to growth on glucose (•) ($\rho = -0.8$, p<0.0001, Spearman rank correlation, significance level $\alpha = 0.0005$) and sucrose (o) ($\rho = -0.5$, p<0.05, Spearman rank correlation, significance level $\alpha = 0.0025$). Efficiency units are conidia generated per molecule of saccharide, growth rate is calculated from the Malthusian growth parameter (mean ± s.e.m., n = 5) which were controlled by varying uptake rates by culturing on varying resource concentrations (1, 0.5, 0.125 and 0.03125 % w/v). Lines (solid = glucose, dashed = sucrose) represent a fit to data of a trade-off geometry directly inferred from the biophysical mechanisms that cause trade-offs **Meyer et al., 2015** (Materials and methods). Typical parameter estimates can be seen in **Figure 3—source data 1**.

The following source data is available for figure 3:

**Source data 1.** Typical parameter estimates obtained by fitting the geometric form of the rate-efficiency trade-off **Meyer et al., 2015** to data in **Figure 3** of the main text.

second social dilemma is key to observing such an outcome. Namely, by removing the self-restraint constraint from the model, we recover the initially expected result, whereby populations consisting entirely of producers are fitter than populations containing a mixture of producers and non-producers (*Harrison et al., 2006*; *Rumbaugh et al., 2009*; *Pollitt et al., 2014*; *Buckling and Brockhurst, 2008*) (*Figure 4—figure supplement 1c*). The same outcome was achieved by neutralising the influence of the rate-efficiency trade-off in spatially homogeneous environments where resources are shared equally between competitors, thus preventing the formation of glucose spikes around producers (*Figure 4—figure supplement 1d*).

To test the mathematical predictions of our model experimentally, we used an in vitro environment in which spatial structure and resource use efficiency could be manipulated. We found that when *M. oryzae* was grown in spatially structured environments containing sucrose as the sole sugar in concentrations for which a rate-efficiency trade-off is effective (1%), the pathogen population fitness was maximised when it contained a mixture of producers and non-producers (*Figure 4a*). This is consistent with both infection observations (*Figure 2a*) and predictions of the model (*Figure 4—figure supplement 1b*). The result also suggests that enhanced fitness of mixed infections is not caused by compensatory up-regulation of genes in non-producers, targeting other host nutrients. In a sucrose concentration (0.01%) where the growth rate is relatively low and so the rate-efficiency trade-off will be weak or non-existent (*Figure 3*), we observed that the pathogen fitness was no longer amplified when non-producers were present in the population (*Figure 4b*), as predicted again by our model (*Figure 4—figure supplement 1c*). The spatial structure of the fungus can be restricted through resource homogenization by supplying resources in the form of glucose (1% w/v) instead of sucrose and hence preventing glucose spikes around producers to occur, or through liquid culture in sucrose-containing growth medium (1% w/v). Consistent with the model predictions, the fitness advantage of mixed populations was lost in terms of biomass production (liquid cultures, *Figure 4c*) and conidia production (resource homogenization, *Figure 4d*). Since invertase production in *M. oryzae* is sucrose-induced (*Figure 1c*) producers do not pay the cost of invertase production and therefore do not suffer a reduction in fitness compared to non-producers in glucose environments (*Figure 4d*).

We conclude that a rate-efficiency trade-off enhanced by hexose availability and spatial structure is the essential pre-requisite for mixed populations of *M. oryzae* to show enhanced fitness and virulence.

## Discussion

Our study suggests that infections comprised purely of highly virulent micro-organisms may be limited in their population fitness by inefficient resource use, through a rate-efficiency trade-off. Introduction of a lower virulence strain can therefore alleviate the constraints on efficiency of resource utilization experienced by the virulent strain, by reducing the local resource concentration. This makes the overall pathogen population more virulent, rather than less virulent as was expected. While a mixture of public good cooperators and cheats has been found to maximize microbial fitness in vitro (*Lee et al., 2010*; *MacLean et al., 2010*), our study shows for the first time that this can be observed in a host-pathogen infection system. A major consequence of this observation is that any competitive displacement virulence reduction strategy that targets the secreted products of pathogens (*Gerding et al., 2015*; *Cotty, 1990*; *Cotty and Bayman, 1993*; *Korolev et al., 2014*; *Jansen et al., 2015*; *Brown et al., 2009*; *Crespi et al., 2014*; *Foster, 2005*) can fail due to complex multi-level interactions between strains that individually show different levels of virulence. It is therefore critical to define and understand cooperative and competitive interactions that must occur within heterogeneous pathogen populations when designing such virulence reduction strategies. This is specifically relevant for systems where pathogens secrete virulence factors into the public domain making them exploitable by non-secretors, rather than delivering virulence factors into individual host cells.

Our findings were derived using infection studies with a synthetic community of the well-studied plant pathogenic fungus *M. oryzae* and the tractability of the rice-fungal interaction enabled us to identify a potential mechanism that could explain the unexpected result. Based on previous research (*MacLean et al., 2010*; *Brown and Taylor, 2009*), we postulated that the interactions between two social traits, public goods production and self-restraint, may collectively increase virulence of the

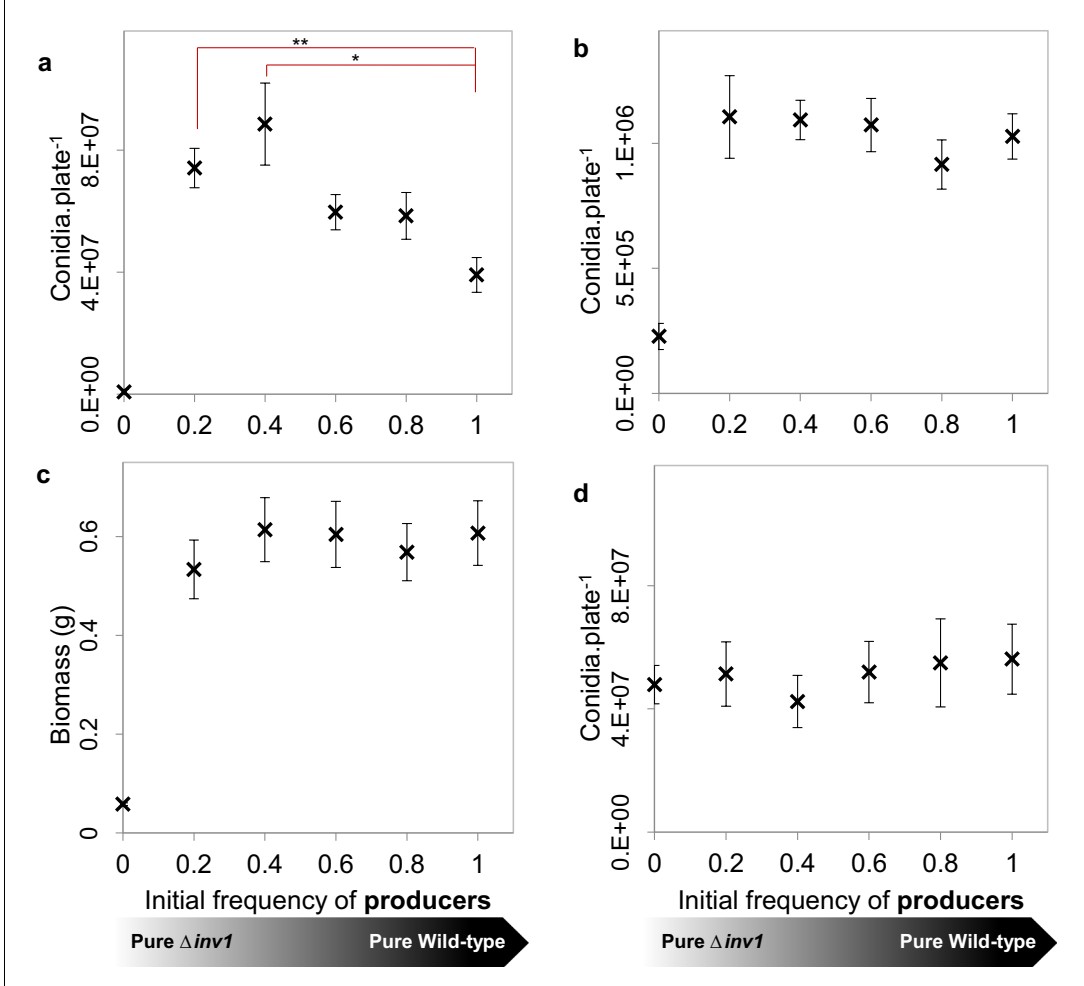

**Figure 4.** Population fitness of INV1 producing Guy11 and the Δ*inv1* mutant in axenic and mixed-strain populations of intermediate frequencies. (a, b, d) Populations were established by inoculation with $1 \times 10^5$ conidia with varying initial frequencies of invertase producers and non-producers, with population fitness being assessed by the number of conidia recovered per plate. (a), on 1% sucrose agar media (mean ± s.e.m., n = 9). Single genotype populations of Guy11 produced more conidia than the Δ*inv1* mutant (p<0.0002, two-sided 2-sample t-test, n = 9). In addition, as when analyzing the in vivo data in *Figure 2a*, we hypothesize that the number of conidia recovered per mixed Guy11 and Δ*inv1* populations is not higher than the number of conidia recovered from Guy11-only populations. However, we can reject this hypothesis using properties of Boolean algebra (*p<0.025, **p<0.004, two-sided 2-sample t-test with Bonferroni correction, n = 9, see Appendix 1A for detailed analysis). (b) Population fitness on 0.01% sucrose agar media to remove the influence of a rate-efficiency trade-off. In this case there was no significant difference amongst fitnesses of mixed populations of producers and non-producers and single genotype populations of producers (p>0.75, $F_{(4, 40)}$ =0.48, one-way ANOVA, n = 9, detecting effect size of 0.79 with the probability of Type II error of 0.01). (c) Population fitness in 1% sucrose liquid media to minimise population spatial structure. There was no significant difference amongst fitnesses of mixed populations of producers and non-producers and pure producer populations (p>0.85, $F_{(4, 40)}$ =0.33, one-way ANOVA, n = 9, detecting effect size of 0.62 with the probability of Type II error of 0.01). Cultures were prepared using a mycelial homogenate and fitness measured as biomass production (dry weight). (d) Population fitness on 1% glucose agar media to remove the need for invertase mediated metabolism and hence spatial heterogeneity in hexoses. There was no significant difference amongst fitnesses of mixed populations of producers and non-producers and pure producer populations (p>0.9, $F_{(4, 40)}$ = 0.24, one-way ANOVA, n = 9, detecting effect size of 0.55 with the probability of Type II error of 0.01).

The following figure supplement is available for figure 4:

**Figure supplement 1.** The interactions between two social traits: public goods production and self-restraint; theoretical results.

pathogen population. Our mathematical model and the subsequent experimental verifications are all consistent with this idea, making multi-trait interactions a credible mechanism to explain the failure of a virulence reduction strategy, based on the introduction of a reduced virulence mutant into a pathogen population.

Due to the inherent complexity of in vivo infection systems, there may of course be other mechanisms at play. One can, for example, consider a hypothetical alternative mechanism behind the amplified virulence we observed in mixed strain infections. While the invertase mutant is less virulent alone (*Figure 2b*), the mutant might be able to use the metabolic energy saved from not producing invertase to increase its relative fitness, for example to fuel production of alternative virulence factors when co-infecting with the invertase-producing strain. However, this is very challenging to verify experimentally. Contrary to this scenario, the mechanism we propose has been successfully tested in vitro (*Figure 4*). Specifically, we recover the synergy between invertase producers and non-producers when there is an interaction between public good production and a rate-efficiency trade off (*Figure 4a*). When we removed the interaction between these traits, this synergy was lost (*Figure 4b–d*).

Strikingly, the mechanism we identified is not unique to our fungal pathogen system and may therefore have wide potential applicability for disease management, because multi-trait cooperative interactions are present in both fungal and bacterial infection. For instance, the vast majority of fungi are reliant upon feeding by secreting public goods (*James et al., 2006*; *Richards and Talbot, 2013*). This, coupled with the ubiquity of rate-efficiency tradeoffs (*Meyer et al., 2015*; *Pfeiffer et al., 2001*; *Beardmore et al., 2011*), means that most fungal pathogens are expected to face the same multi-level social interactions as *M. oryzae*. Moreover, pathogenic bacteria employ multiple social traits such as secretion of extracellular signaling molecules responsible for the production and secretion of virulence related proteins (*Pirhonen et al., 1993*; *Lee and Lee, 2010*). Indeed, some bacterial populations containing a mixture of public good producers and non-producers have been found to grow better in vitro than pure producer populations (*Lee et al., 2010*).

In addition, our findings are potentially relevant to newly proposed cancer treatments. Tumour cells produce public goods in the form of secreted growth factors or angiogenic factors to gain access to nutrients. Treatments that promote non-producing cheats or deploy anti-angiogenic drugs have been proposed as a strategy to weaken tumours (*Korolev et al., 2014*; *Jansen et al., 2015*). However, tumour cells are likely to experience a tragedy of the commons through rapid glucose metabolism (*Gillies and Gatenby, 2007*) - given the biophysical necessity of rate-efficiency trade-offs (*Meyer et al., 2015*). Therefore, suggested treatments that disrupt cooperation and promote the evolution of non-cooperators (*Korolev et al., 2014*; *Jansen et al., 2015*) might result in an increase in tumour fitness, analogous to the observations in this study, leading to devastating consequences for the host.

Finally, in light of the antibiotic resistance crisis, there is a growing interest in synthetic biology therapies for the treatment of infections (*Ruder et al., 2011*). Engineering viruses that lower pathogen virulence, for example, is being investigated for control of plant diseases (*Nuss, 2005*; *Kanhayuwa et al., 2015*) and was recently developed in vitro for bacterial pathogens (*Lu and Collins, 2007*). Similarly, synthetic low-virulence strains have been used in animal infection models (*Harrison et al., 2006*; *Rumbaugh et al., 2009*; *Pollitt et al., 2014*) and suggested as a treatment of plant pathogens (*Frey et al., 1994*; *Cleveland et al., 1990*). Moreover, with the development of genome editing technologies like CRISPR/Cas9, the creation of low virulence strains to treat disease could very soon be a feasible strategy for a wide range of systems. This study provides a mechanistic foundation for the analysis of why such disease control strategies may fail unless social interactions between micro-organisms are fully considered.

## Materials and methods

### Fungal strains, growth conditions, and DNA analysis

Strains of *Magnaporthe oryzae* in this study are derived from the wild-type Guy-11 strain (*Leung et al., 1990*) and the subsequently generated GFP expressing (*ToxAp:SGFP*) strain (*Sesma and Osbourn, 2004*). Typical procedures for fungal growth, maintenance, transformation and DNA extraction were performed as previously described (*Talbot et al., 1993*) with nucleic acid

assessment and manipulation performed according to standard practices (*Sambrook et al., 1989*). Nucleotide sequences were obtained from the *Magnaporthe oryzae comparative* Sequencing Project, Broad Institute of Harvard and MIT (http://www.broadinstitute.org/ RRID:SCR_007073). Primer nucleotide sequences used in this study can be found in *Supplementary file 1*. Southern blot analysis was performed using digoxigenin(DIG)-labelled (Roche Applied Science) probes visualised with CDP -*Star* Chemiluminescent Substrate (Sigma Aldrich). Sucrose used throughout the experiments was from 30% (w/v) stock solution, filter sterilised with 1 mM Tris/HCl, pH 8, to inhibit acid-catalysed autohydrolysis. In vitro experiments with growth on sucrose were performed in 1% (w/v) (except in cases where expressed otherwise for model validation), which is representative of reported sucrose concentrations found within rice leaves (*Dallagnol et al., 2013*).

## Measurement of fitness

Fitness was calculated from Malthusian growth parameters (m) as described previously (*Lenski et al., 1991*), where:

$m = \ln [N(1)/N(0)]/d$, when $N(1)$ = final density, $N(0)$ = initial density, and $d$ = time.

During *in planta* fitness measurements, because some replicates returned zero values, violating the assumption of exponential growth in m, we employed a relative fitness (v) that compares changes in the relative frequencies (*Ross-Gillespie et al., 2007*), where: $v = x_2 (1 - x_1) / x_1 (1 - x_2)$, when $x_1$ = initial frequency and $x_2$ = finial frequency.

Population fitness and strain frequency were measured by spore production, except for in liquid cultures where dry-weight biomass was measured, as liquid is not conducive to conidiogenesis (*Zhang et al., 2014*). Both traits contribute to the ability to survive and reproduce, so represent appropriate measures of fitness (*Pringle and Taylor, 2002*). Importantly for pathogenic fungi, sporulation permits transmission to new hosts. Spore production quantification has been used as a direct measure of fungal reproduction and transmissibility (and hence fitness), which is thought to correlate with the degree of host exploitation and resource uptake (*López-Villavicencio et al., 2011*). Measurements were taken at specific time points to allow resources to be used up so that population fitness is based on final population size. This time varies between experiments, as detailed below, depending on the resource concentration and environment employed.

## Generating a less virulent strain of *M. oryzae*

An invertase non-producing mutant, $\Delta inv1$, was generated in a *ToxA:SGFP* (*Sesma and Osbourn, 2004*) background by the Split-Marker technique (*Kershaw and Talbot, 2009*). Targeted gene replacement was achieved by replacing the putative *INV1* invertase gene with a 2.8 kb sulfonylurea resistance allele (*ILV1*). In a first round of PCR, a 1 kb genomic fragment upstream (LF) and downstream (RF) of the ORF were amplified. Separately, 1.6 kb overlapping fragments of the 5' and 3' end of *ILV1* were amplified. Amplicons produced were fused in a second round of PCR to produce two larger fragments of 2.6 kb. The constructs were used to transform *M. oryzae* and gene replacement achieved by homologous recombination (*Figure 1—figure supplement 1a*). Gene replacements were confirmed using digoxigenin (DIG)-labelled Southern blot analysis by fragment size differentiation, following restriction endonuclease digestion of genomic DNA with *Xho*I. Blots were probed with the left flank region (LF, 1 kb upstream of the ORF). Analysis showed a 1.4 kb size difference with wild-type genotype band at 4.2 kb with mutants at 5.6 kb (*Figure 1—figure supplement 1b*).

## Phenotypic assessment

Invertase activity was tested based on a spectrophotometric stop reaction where acid hydrazide generates yellow anions by reacting with reducing carbohydrates (glucose and fructose) in alkaline solutions (*Bacon, 1955*, *Lever et al., 1972*). Secreted invertase was measured from induction media comprised of the filtrate of fungal cultures in minimal media (*Talbot et al., 1997*) with glucose and/or sucrose (10 g.L$^{-1}$). Biomass was first established for 48 hr in 50 mL complete media (*Talbot et al., 1993*) (CM) inoculated with a fragmented 5 cm$^2$ plug from a CM agar plate. Biomass was harvested, washed with sterile deionised water (sdH$_2$O), drained and then transferred to the induction media for 18 hr. The filtrate was extracted and snap frozen in liquid N$_2$ prior to being lyophilized. Desiccated samples were rehydrated with 2 mL sdH$_2$O and 1 mL dialysed at 4°C with 10 K MWCO

Snakeskin Dialysis tubing (Thermo Scientific), against 5 L 10 mM sodium acetate buffer (pH 5) that was replaced fresh once during the 24 hr dialysis period. Invertase activity was measured by combining 100 µL of the dialysed sample with 900 µL 29.2 mM sucrose substrate (1% w/v in 100 mM sodium acetate buffer, pH 4.5) at 55°C for 20 min. One hundred microliters of this mixture was then transferred to 2.9 mL 0.5% (w/v) PAHBAH (p-Hydroxybenzhydrazide in 0.5 M NaOH). Reaction mixtures were heated at 100°C for 5 min, then cooled to room temperature and absorbance read at 410 nm. Measurements were made against un-inoculated blanks; with monosaccharides liberated being quantified by comparison to a standard curve generated with a glucose dilution series.

Biomass production was assessed by inoculating a 5 cm$^2$ plug of *M. oryzae* mycelium, from a CM agar plate at the periphery of an actively growing culture, into 150 mL liquid CM and blended. Mycelium from these were extracted after 48 hr, washed with sdH$_2$O, drained and 1 g wet weight (=0.0635 g dry weight ± 0.00468 s.e.m.) transferred to 150 mL liquid minimal medium (*Talbot et al., 1997*) (MM) with sucrose (10 g.L$^{-1}$) replacing glucose. Fungal biomass was harvested after 120 hr and dry weight established.

Sporulation was quantified in vitro from 25 mL MM (+ glucose/sucrose) agar plates. Conidia were harvested after 12 d by flooding with sdH$_2$O and agitating the culture surface. The liquid was then filtered to remove debris and conidia enumerated with a haemocytometer.

Growth rate measurements used in *Figure 3* were calculated from Malthusian growth parameters (*Lenski et al., 1991*) which were controlled by varying uptake rates by culturing on varying resource concentrations (1, 0.5, 0.125 and 0.03125% w/v) (*Weusthuis et al., 1994*). For the same figure, efficiency was calculated as number of conidia produced per molecule of glucose or sucrose.

## Functional complementation of the *M. oryzae* Δ*inv1* mutant

Functional complementation was achieved by PCR cloning the *INV1* ORF, with 1.8 kb upstream and 0.5 kb downstream to incorporate the native promoter and terminator sequences, into Strataclone (Stratagene) as an *Eco*RI/*Hind*III fragment,. To this vector, the *BAR* gene conferring bialophos (BASTA) resistance was ligated as a *Not*I/*Spe*I fragment. The resulting vector was transformed into the Δ*inv1* mutant and transformants assessed for single integration of the *INV1* gene by Southern blot analysis with digestion of genomic DNA with *Ahd*I (data not shown), and restored wild-type growth morphology on sucrose media (*Figure 1a*).

## Functional complementation of a yeast invertase mutant

The strains of *Saccharomyces cerevisiae* used were kindly supplied by the Botstein lab (Princeton University, USA) and the Fink lab (Whitehead Institute, USA). DBY1034 has the genotype *MATa his4-539 lys2-801 ura3-52 SUC2*, DBY1701 is a *SUC2* deletion strain with genotype *MATa his4-539 lys2-801 ura3-52 suc2Δ9* (*Kaiser and Botstein, 1986*). The expression vector was constructed by modification of the NEV-E vector (*Sauer and Stolz, 1994*). The plasma membrane ATPase gene promoter (P$_{PMA1}$) was substituted for the constitutive promoter of the glyceraldehyde-3-phosphate dehydrogenase gene (P$_{GPD}$). Complementation analysis was performed by cloning the *M. oryzae* ORF into the resulting vector and then transformed into DBY1701, with the empty vector transformed into DBY1701 and DBY1034 as a negative and positive control respectively. Growth rates were assessed in 640 µL cultures of 25 mM sucrose, 5 g.L$^{-1}$ ammonium sulfate, 1.7 g.L$^{-1}$ yeast nitrogen base w/o amino acids or ammonium sulfate, 50 mg.L$^{-1}$ L-lysine and 20 mg.L$^{-1}$ L-histidine, in 48-well suspension culture plates (Cellstar Greiner Bio-One) with 700 r.p.m. orbital shaking. Optical density measurements were made at 620 nm in a FLUOstar Omega microplate reader (BMG Labtech) and converted to CFU using a calibration curve of known densities.

## Confirmation of invertase activity as a public good in populations of *M. oryzae*

Mycelium from Guy11 grown in CM was harvested, washed with sdH$_2$O and 1 g wet weight, inoculated into 150 mL MM + sucrose for 48 hr. Fungal biomass was extracted by filtration and the filtrate, containing generated public-goods such as invertase, was re-supplied with MM + sucrose nutrients before being inoculated with the Δ*inv1* mutant of *M. oryzae*. Biomass generation was measured as dry weight of Δ*inv1* after 72 hr. The initial Guy11 was replaced with Δ*inv1* as a negative control.

## *M. oryzae* growth competition assays

The strains were distinguishable due to the defector strain being tagged with GFP (sGFP) and the co-operator strain being tagged with RFP (3mCherry), both being driven by a constitutively expressed promoter (*ToxA*) (*Lorang et al., 2001*) that was selectively neutral (*Figure 2—figure supplement 2*).

Mixed-strain competition experiments were performed, as described previously for *S. cerevisiae*, to establish degrees of population structure (*MacLean et al., 2010*), but with the following modifications. Conidia from 10–12 d CM agar plates were harvested and washed with sdH$_2$O and resuspended in semi-solid (2 g.L$^{-1}$ agar) MM-C (*Talbot et al., 1997*) to a concentration of $2.5\times10^5$.mL$^{-1}$. Conidia were then inoculated onto 25 mL MM + sucrose (10 g.L$^{-1}$ or 0.1 g.L$^{-1}$) agar plates with $1\times10^5$.plate$^{-1}$ in a 4×5 array with patch midpoints separated by 12 mm. Data in *Figure 1e* and *Figure 4a,b,d* used the 'unstructured' configuration as in *MacLean et al. (2010)*, whereas *Figure 1—figure supplement 3* used the 'structured' configuration as in *MacLean et al. (2010)*. Competition cultures were incubated for 12 d, conidia collected and counted as described above, with wild type and Δ*inv1* mutants identified by epifluorescence microscopy (Leica M205FA).

To minimise the influence of population structure and hence spikes in glucose concentrations where the rate-efficiency trade off would peak, competitions were performed in liquid sucrose (10 g. L$^{-1}$) minimal media at 125 r.p.m. with strains inoculated as wet biomass established from CM, as described above, totaling 1 g in strain proportions of 0, 20, 40, 60, 80 or 100%. Fitness was measured as dry weight biomass established. Growth curves were established for each proportion of strains to establish the point at which resources had been consumed and biomass peaked. This was 5 d for all combinations except 20% producer which required 9 d, and axenic non-producer populations failed to increase in biomass. In case biomass did not capture the subtleness of the synergistic effect, we also eliminated the heterogeneous glucose spikes that are present in structured conditions when sucrose breakdown by invertase is required. This was achieved by performing the same experiment as the in vitro set up that captured the synergy between strains (*Figure 4a*) but in a nutritional environment that does not require external digestion (1% (w/v) glucose, *Figure 4d*).

The RFP expression vector was constructed by ligation of a triple tandem repeated mCherry *NcoI/NdeI* fragment, *ToxA* promoter *SacII/NcoI* fragment, and *trpC* terminator *NdeI/XhoI* fragment, into the transformation vector pCB1532 conferring sulfonylurea resistance. Transformants were assessed for positive fluorescence signal by epifluorescence microscopy (Olympus IX81).

## Pathogenicity and *in planta* fitness assay of *M. oryzae*

We used a quantitative localised leaf spot infection assay to assess pathogen fitness and disease virulence so that the exact number of conidia in a specific area and the subsequent fitness of the pathogen could be determined, in addition to permitting intimate interactions between individuals of the infecting population (*Figure 2*). Each treatment was inoculated onto an individual rice plant and only a single leaf per plant was infected (*Supplementary file 2*). This was performed according to a previously described protocol (*Jia et al., 2003*), with the following modifications. Leaves of 21-day-old seedlings of rice cultivar CO39 were inoculated using intact seedlings because detached leaves may trigger defence responses and inhibit sink induction at infection sites and thus nutrient acquisition by the pathogen. Each disease patch was initiated by inoculation with a 20 µL suspension of $5\times10^4$ conidia.mL$^{-1}$ in 0.2% (w/v) gelatine. The inoculum concentration was chosen because it was sufficient to facilitate the full disease cycle to be completed, such that the ability to cause disease could be accurately measured based on sporulation from the disease lesion. The inoculum level resembles that which would occur under disease epidemic conditions, as each disease lesion typically generates 20,000–50,000 spores per day in severe rice blast infections (*Wilson and Talbot, 2009*). Infection proceeded for 7 d with droplets blotted after the initial 26 hr. Images of the infection lesions used for virulence quantification (*Figure 2b*, *Figure 2—figure supplement 1*) were captured at this time using an Epson Expression 1680 Pro scanner (1200 d.p.i.). To quantify pathogen fitness by conidia production, the lesions were excised from leaves and placed under high humidity to induce sporulation for 48 hr. Images of sporulating lesions in *Figure 2d* were captured at this time by epifluorescence microscopy (Leica M205FA, processed using ImageJ, National institutes of Health, USA).

Fourteen infections per treatment were assessed for conidia production by the pathogen. Conidia were extracted by flooding lesions with 200 µL sdH$_2$O, vortexed and lesion surfaces gently abraded,

before enumeration with a haemocytometer. This protocol was repeated three times giving a total number of n = 42 replicates (3 (repeated protocol) x 14 (infections enumerated) =42) per each inoculum condition (*Figure 2a*). Note that as we consider six inoculum conditions with varying frequencies of competitors, a total of 252 disease lesions (6 (conditions) x 42 (replicates) =252) were assessed for conidiation.

Disease virulence in terms of lesion area was quantified using image analysis software (ImageJ, National Institutes of Health, USA). As we consider three inoculum conditions with varying frequency of competitors and n = 20 replicates for each condition, a total of 60 disease lesions were assessed for size of the lesion area (*Figure 2b* and *Figure 2—figure supplement 1*).

Mixed infections were assessed by epifluorescence microscopy to observe co-infecting Guy11 (RFP) and Δ*inv1* (GFP) invading neighbouring / same host plant cells where invertase exploitation is more likely to occur (*Figure 2c*). This was performed using leaf sheath inoculation assays based on those previously described (*Kankanala et al., 2007*), using a conidial suspension of $5 \times 10^4$ .mL$^{-1}$ in 0.2% (w/v) gelatine, and cell invasion observed, after dissection, by light microscopy (Olympus IX81).

## Data analysis

Statistical tests were performed using R version 3.1.1 and Statistics and Machine Learning Toolbox in MATLAB R2015a.

Pairwise comparisons of invertase activity, conidia production, biomass and disease lesion size between two populations (*x* and *y*) were made by 2-sample t-test, if approximate normality and F-test for equal variance were satisfied. When the assumptions were violated, non-parametric Mann-Whitney U or t-test for unequal variances (Behren's Fisher problem) was used. In all cases the null hypothesis was of the same format: invertase activity/conidia production/biomass/disease lesions of population *x* is not different to that of population *y*.

Multiple comparisons of different inoculum conditions were conducted using the following test. If the data violated assumptions of ANOVA, the non-parametric Kruskal Wallis test was performed, followed by two-sided Mann-Whitney U test with Bonferroni correction (in vivo data *Figure 2a*). Otherwise one-way ANOVA was used followed by two-sided t-test with Bonferroni correction (in vitro data *Figure 4* and in vivo data *Figure 2—figure supplement 4*); for full details see Appendix 1A. Note that for testing the assumptions of one-way ANOVA, residuals were observed to be approximately normally distributed by plotting a Normal Q-Q plot and heteroscedasticity measured by Fligner-Killen test of homogeneity.

The data in *Figure 2—figure supplement 4* shows a log transformation of the data from *Figure 2a*. This transformation was carried out to improve the normality of the data as per previous studies (*Köhler et al., 2009a*; *Lu and Collins, 2007*; *Köhler et al., 2009b*), as well as to satisfy homogeneity of variance so that the appropriate parametric tests could be applied. Departures from normality were monitored by skewness and kurtosis with 'substantial' departures considered according to (*West et al., 1995*). The multiple comparisons of different inoculum conditions of the log transformed data in *Figure 2—figure supplement 4* were conducted using one-way ANOVA, followed by two-sided t-test with Bonferroni correction (for full details of the procedure see Appendix 1A).

Data fitting was performed using non-linear fitting routines in MATLAB or in R.

For in vitro data, *post-hoc* power analysis was performed when the null hypothesis of equal means could not be rejected. This was done in R using Package *pwr* version 1.1–3 with significance level set at $\alpha = 0.05$.

The sample sizes for in vivo data were chosen to maximize the number of infections that could be achieved in the plant growth room while minimizing the chances of cross-infection.

Performing an *a priori* power analysis was not appropriate for our study, as we were not seeking to detect pre-specified differences between samples.

## Fitting geometric trade-offs to data

Recent paper (*Meyer et al., 2015*) inferred the geometry of the rate-efficiency trade-off, directly from the biophysical mechanisms that cause it. The resulting five parameter geometric trade-off has the form:

$$(growth\ rate,\ efficiency) = \left( c(H)\frac{HV_{max}}{K_m + H}, c(H) \right),$$

where $H$ is the resource concentration, $V_{max}$ is the maximal rate of sugar uptake while $K_m$ denotes sugar half saturation constant (see *Meyer et al., 2015* for more details).

The resource efficiency is a function of sugar and takes the form as in *Meyer et al. (2015)*

$$c(H) = c_{hi}\frac{1}{1 + pH} + \frac{pH}{1 + pH}c_{lo}.$$

The parameter $c_{hi}$ represents the highest spore number per molecule of resources attainable, achieved at the lowest sugar concentrations, whereas $c_{lo}$ is the spore numbers attained when sugar is abundant, $p$ is a phenotype that controls the rate of decrease in efficiency with increasing sugar supply.

We first fit $c_{hi}$, $c_{lo}$ and $p$ to the data on efficiency as a function of sugar, then we fit $V_{max}$ and $K_m$ to the growth rate data as a function of sugar, both steps using non-linear fitting routines in MAT-LAB. The resulting rate-efficiency data fit is shown in *Figure 3* and typical parameter estimates are given in *Figure 3—source data 1*.

In addition, negative monotonic correlation between growth rate and efficiency data was verified using Spearman's rank correlation test.

## Acknowledgements

We would like to thank Robert Beardmore and James Cresswell for helpful discussions.

## Additional information

### Funding

| Funder | Grant reference number | Author |
| --- | --- | --- |
| Natural Environment Research Council | NE/E013007/3 | Richard J Lindsay Ivana Gudelj |
| Natural Environment Research Council | Doctoral training grant studentship | Richard J Lindsay Ivana Gudelj |
| Engineering and Physical Sciences Research Council | Doctoral training grant studentship | Bogna J Pawlowska Ivana Gudelj |
| European Research Council | no. 294702 GENBLAST | Nicholas J Talbot Ivana Gudelj |
| European Research Council | no. 647292 MathModExp | Ivana Gudelj Nicholas J Talbot |

The funders had no role in study design, data collection and interpretation, or the decision to submit the work for publication.

### Author contributions

RJL, Designed the experiments, conducted the experiments, wrote the manuscript; MJK, Designed the experiments, conducted the experiments; BJP, Developed the mathematical model, conducted numerical simulations; NJT, Designed the experiments, wrote the manuscript; IG, Conceived the idea, designed the experiments, developed the mathematical model, wrote the manuscript

### Author ORCIDs

Nicholas J Talbot, http://orcid.org/0000-0001-6434-7757
Ivana Gudelj, http://orcid.org/0000-0003-3450-6854

## Additional files

**Supplementary files**
• Supplementary file 1. Primers used in this study.
• Supplementary file 2. Infection assay protocol

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

## Appendix

## A Data analysis

In this section we describe the statistical procedure based on Boolean logic design, used to analyse in vivo infection data presented in *Figure 2a* of the main text and in vitro data presented in *Figure 4* of the main text.

### A.1 Notation and basic operations of Boolean algebra

A Boolean algebra has the following operators:

$$\wedge \ \ (\text{denoting AND}),$$
$$\vee \ \ (\text{denoting OR}) \text{ and}$$
$$\neg \ \ (\text{denoting NOT}).$$

We define a *proposition* to be a statement which is either *true* or *false*. If $A_1$ and $A_2$ denote two propositions then the truth tables for the three operators $\wedge$, $\vee$ and $\neg$ are:

| $A_1$ | $A_2$ | $A_1 \vee A_2$ |
|---|---|---|
| true | true | true |
| true | false | true |
| false | true | false |
| false | false | false |

| $A_1$ | $A_2$ | $A_1 \wedge A_2$ |
|---|---|---|
| true | true | true |
| true | false | true |
| false | true | false |
| false | false | false |

| $A_1$ | $\neg A_2$ |
|---|---|
| true | false |
| false | true |

The above truth tables can be extended to any number of propositions as the next Example 1 shows.

Example 1. *Let* $\{A_i | i = 1..n, n \in \mathbb{N}\}$ *denote a set of n propositions. Then,*

$$(A_1 \vee A_1 \vee ... \vee A_n) \text{ is true}$$

*if there exist at* **least one** $i \in \{1, 2..n\}$ *for which* $A_i$ *is true. Similarly,*

$$(A_1 \wedge A_1 \wedge ... \wedge A_n) \text{ is true}$$

*if* $A_i$ *is true* **for all** $i \in \{1, 2..n\}$.

The operators $\wedge$, $\vee$ are associative as explained in the next theorem.

**Theorem** (Associative law). *Let* $A_1$, $A_2$, $A_3$ *denote three propositions. Then*

$$(A_1 \vee A_2) \vee A_3 \ = A_1 \vee (A_2 \vee A_3)$$
$$(A_1 \wedge A_2) \wedge A_3 \ = A_1 \wedge (A_2 \wedge A_3)$$

Before we present an example on how to negate inequalities (Example 2), we state the following theorem.

**Theorem** (Trichotomy law for real numbers). *For all* $a, b \in \mathbb{R}$, one and only one of the following holds: $a < b$, $a > b$ or $a = b$.

Example 2. *Let* $A$ *denote the proposition:* $a \leq b$, *where* $a, b \in \mathbb{R}$. *Then, due to the Trichotomy Law for real numbers*

$$\neg A = \neg(a \leq b) = (a > b).$$

Finally, we state a theorem widely used in Boolean logic design.

**Theorem** (De Morgan's laws). *Let $A_1$ and $A_2$ denote two propositions. Then*

$$\neg(A_1 \wedge A_2) = \neg A_1 \vee \neg A_2$$
$$\neg(A_1 \vee A_2) = \neg A_1 \wedge \neg A_2$$

Note, since the operators $\wedge$ and $\vee$ are associative, De Morgan's laws hold for *any* number of propositions as illustrated in Example 3.

Example 3. *Let $\{A_i | i = 1..n, n \in \mathbb{N}\}$ denote a set of $n$ propositions. Then*

$$
\begin{aligned}
\neg(A_1 \wedge A_2 \wedge ... \wedge A_n) &= \neg A_1 \vee \neg(A_2 \wedge ... \wedge A_n) \\
&= \neg A_1 \vee \neg A_2 \vee \neg(A_3 \wedge ... \wedge A_n) \\
&... \\
&= \neg A_1 \vee \neg A_2 \vee ... \vee \neg A_n
\end{aligned}
$$

*Similarly,*

$$\neg(A_1 \vee A_2 \vee ... \vee A_n) = \neg A_1 \wedge \neg A_2 \wedge ... \wedge \neg A_n.$$

# A.2 Analysis of in vivo data

We consider two strains of *Magnaporthe oryzae*: the wild type, $Guy11$, which secretes a public good invertase needed to sequester nutrients from its host plant, and an isogenic mutant $\Delta inv1$ exhibiting reduced secreted invertase activity. The $\Delta inv1$ mutant shows dramatically reduced fitness during plant infections when present in isolation (***Figure 2a***) as measured by spore production at the end of an infection cycle and also exhibits lower virulence, measured by the area of disease lesion coverage of an infected leaf (***Figure 2b***, ***Figure 2— figure supplement 1***).

Since we show in the main text that invertase production in *M. oryzae* is a cooperative trait, the invertase producer ($Guy11$ ) is termed a cooperator while the non-producer ($\Delta inv1$) is termed a cheat. Using the data presented in ***Figure 2a***, we ask the following question: can an introduction of a public good non-producer (cheat) into a population of public goods producers (cooperators), lead to an increase in the overall population fitness? According the existing social theory (***Hamilton, 1964a***, ***1964b***), the answer to the above question is no, so we formulate the following hypothesis:

$H_0$: *the fitness of mixed populations of cooperators ($Guy11$) and cheats ($\Delta inv1$) is not higher than the fitness of cooperator-only populations.*

Mathematically, $H_0$, can be written in the following way. Let $W_{XGuy11} \in \mathbb{R}$ denote the fitness of $XGuy11$ population, where $X \in \{20, 40, 60, 80, 100\}$ represents the percentage of $Guy11$ strain in the initial infection inoculum. For example, $20Guy11$ denotes a population consisting of 20% $Guy11$ cooperators and 80% $\Delta inv1$ cheats. Subsequently, using the Trichotomy law for real numbers and Boolean algebra, $H_0$ can be written as

$$
\begin{aligned}
H_0 := \ &\{(W_{20Guy11} \leq W_{100Guy11}) \wedge (W_{40Guy11} \leq W_{100Guy11}) \\
&\wedge (W_{60Guy11} \leq W_{100Guy11}) \wedge (W_{80Guy11} \leq W_{100Guy11})\}.
\end{aligned}
\tag{1}
$$

From De Morgan's laws

$$\neg H_0 := \{\neg(W_{20Guy11} \leq W_{100Guy11}) \vee \neg(W_{40Guy11} \leq W_{100Guy11}) \\ \vee \neg(W_{60Guy11} \leq W_{100Guy11}) \vee \neg(W_{80Guy11} \leq W_{100Guy11})\}, \tag{2}$$

and we further apply Trichotomy law for real numbers (as illustrated in Example 2) to re-write (2) as

$$\neg H_0 := \{(W_{20Guy11} > W_{100Guy11}) \vee (W_{40Guy11} > W_{100Guy11}) \\ \vee (W_{60Guy11} > W_{100Guy11}) \vee (W_{80Guy11} > W_{100Guy11})\}. \tag{3}$$

## A.2.1 Application of Boolean algebra to in vivo data: combining logic and probability

We can reject hypothesis $H_0$ if we can show that $\neg H_0$ is true. To this end, using the in vivo infection data in **Figure 2a** of the main text, we first show that there is significant difference among $W_{20Guy11}$, $W_{40Guy11}$, $W_{60Guy11}$, $W_{80Guy11}$ and $W_{100Guy11}$ ($p < 0.0001$, Kruskal Wallis test).

Next, to show that $\neg H_0$ is true, using standard properties of Boolean algebra (as illustrated in Example 1) applied to (**Equation 3**), it is sufficient to show that there exist at least one $X \in \{20, 40, 60, 80\}$ such that $(W_{XGuy11} 100Guy11)$. Therefore we make the following four pairwise comparisons using two-tailed Mann-Whitney U test with Bonferroni correction:

$$(W_{20Guy11} \text{ v.s } W_{100Guy11}), (W_{40Guy11} \text{ v.s } W_{100Guy11}), \\ (W_{60Guy11} \text{ v.s } W_{100Guy11}) \text{ and } (W_{80Guy11} \text{ v.s } W_{100Guy11}), \tag{4}$$

and show that $W_{80Guy11} > W_{100Guy11}$ ($p < 0.0365, W = 1174$, two-tailed Mann-Whitney U test with Bonferroni correction, $n = 42$). Consequently, we conclude that $\neg H_0$ is true and thus $H_0$ can be rejected.

Note that there is nothing 'special' about $80Guy11$ populations. We would have also rejected $H_0$ had we instead found that $(W_{20Guy11} > W_{100Guy11})$ or $(W_{40Guy11} > W_{100Guy11})$ or $(W_{60Guy11} > W_{100Guy11})$. In fact, we reject $H_0$ when tested on the data generated by an equivalent in vitro experiment (**Figure 4a** of the main text), because we find three mixed populations, namely $20Guy11, 40Guy11$ and $60Guy11$ whose fitness is greater than the fitness of $100Guy11$ (see Appendix 1A.3 for more detail).

Importantly, we also show that fitness positively correlates with virulence and that the mixed populations of cooperators and cheats whose fitness is higher than the fitness of cooperator only populations, also have higher virulence, causing significantly more damage to the host than cooperator only infections ($p < 0.004$, two-sided 2-sample t-test with unequal variances, $n = 20$, data shown in **Figure 2b** of the main text).

Our analysis shows that, contrary to the current understanding based on social evolution theory (**Hamilton, 1964a**, **1964b**), animal (**Harrison et al., 2006**; **Rumbaugh et al., 2009**; **Pollitt et al., 2014**; **Köhler et al., 2009**) and plant (**Frey et al., 1994**) models, as well as clinical (**Gerding et al., 2015**) and agricultural field studies (**Cotty and Bayman, 1993**; **Amaike and Keller, 2011**), an introduction of cheats with low virulence into a population of cooperators with high virulence can increase both fitness and virulence of the population. This provides the first experimental evidence that a promising virulence reduction strategy can have devastating consequences for the host.

## A.3 Analysis of in vitro data

We perform a similar statistical procedure to the one described in A.2 but now on the in vitro data presented in *Figure 4* of the main manuscript. This data contributes to the mechanistic understanding behind the unexpected result obtained in *Figure 2a* of the main text.

First we show that in vitro data in *Figure 4a* is consistent with infection observations in *Figure 2a*, in particular that an introduction of cheats into a population of cooperators can lead to an increase in population fitness. Following the same logical argument described in A.2, in order to test the hypothesis $H_0$ defined in (1) we first show that there is significant difference among $W_{20Guy11}$, $W_{40Guy11}$, $W_{60Guy11}$, $W_{80Guy11}$ and $W_{100Guy11}$ ($p<0.003$, $F_{(4,40)} = 74.95$, one-way ANOVA).

Next, to show that $\neg H_0$, we make the four pairwise comparisons defined in (4) using two-tailed 2-sample t-test with Bonferroni correction to check whether there exist at least one $X \in \{20, 40, 60, 80\}$ such that ($W_{XGuy11} > W_{100Guy11}$). Subsequently, we show that $W_{20Guy11} > W_{100Guy11}$($p<0.004$, two sided 2-sample t-test with Bonferroni correction, $n = 9$) and $W_{40Guy11} > W_{100Guy11}$ ($p<0.02$, two sided 2-sample t-test with Bonferroni correction, $n = 9$). Therefore using properties of Boolean algebra we conclude that $\neg H_0$, as represented in (3), is true and thus we can reject $H_0$ for the in vitro data presented in *Figure 4a*. This demonstrates that the in vitro data is consistent with the in vivo infection data in *Figure 2a*.

Next, we turn our attention to the data in *Figure 4b–d* of the main text. Each of these three figures presents the outcome of an experiment in which a particular assumption is removed from the environment in which *Figure 4a* data was obtained. That way we can determine the significance of a removed assumption on the result observed in *Figure 4a*. In particular we (i) remove the rate-efficiency assumption by conducting experiments on agar plates in low sucrose (*Figure 4b*); (ii) minimise population spatial structure by considering liquid shaken environments (*Figure 4c*); and (iii) remove resource spatial structure by conducting experiments on agar plates in glucose media (*Figure 4d*).

We find that there is no significant difference among $W_{20Guy11}$, $W_{40Guy11}$, $W_{60Guy11}$, $W_{80Guy11}$ and $W_{100Guy11}$ for the data in *Figure 4b* ($p>0.75$, $F_{(4,40)} = 0.48$, one-way ANOVA, $n = 9$), *Figure 4c* ($p>0.85$, $F_{(4,40)} = 0.33$, one-way ANOVA, $n = 9$) and *Figure 4d* ($p>0.9$, $F_{(4,40)} = 0.24$, one-way ANOVA, $n = 9$). This means that removal of the rate-efficiency trade-off (*Figure 4b*) or spatial structure (*Figure 4c,d*) from the environment destroys the result observed in *Figure 4a* that an introduction of cheats into a population of cooperators can increase population fitness.

We conclude that a presence of a rate-efficiency trade-off and spatial structure are the essential pre-requisites for mixed populations of cooperators and cheats to show enhanced fitness and virulence.

In general, note that for testing the assumptions of One-way ANOVA, residuals were observed to be approximately normally distributed by plotting a Normal Q-Q plot and heteroscedasticity measured by Fligner Killen test of homogeneity.

## B Multi-trait mathematical model

Our empirical study has shown that *M. oryzae* population fitness, measured as spore production, is maximised when the population contains a mixture of invertase producers and non-producers (*Figure 2a* of the main text). Based on the previous theoretical work (*MacLean et al., 2010*) we postulate that this observation is driven by the presence of two social traits. The first trait is the extracellular production of a public good, invertase, that breaks down a complex sugar (sucrose) into simple sugars (hexose). The second social trait is intracellular self-restraint, and relates to the efficiency with which a cell exploits newly-acquired simple sugars.

To test this, we develop a mathematical model of invertase production in *M. oryzae* with the pathogen growth constrained by a rate-efficiency trade-off. Our model needs to capture key features of our experimental system, but also needs to be sufficiently simple so that we can manipulate and probe the spatio-temporal dynamics of extracellular enzymes, resource concentrations and pathogen population densities. This will enables us to specifically focus on the effects of public goods production and self-restraints on population fitness.

Based on a simplified version of MacLean et al. (*MacLean et al., 2010*) and in line with previous models of invertase production (*Gore et al., 2009*), we make the following assumptions.

## Growth kinetics

In our model we consider two strains, an invertase producer (the wild-type $Guy11$) and a non-producer (the $\Delta inv1$ mutant). Both strains take up resources $R$ and use them to generate ATP using a simple, unbranched pathway (*Pfeiffer et al., 2001*). The rate of ATP production in the pathway is denoted by $J^{ATP}$ and is given by

$$J^{ATP} = \eta^R_{ATP} J^R$$

where $J^R$ denotes the rate of the pathway which is a function of resource concentration $R$ and is mathematically represented by $J^R(R)$. The term $\eta^R_{ATP}$ denotes the number of ATP molecules produced in the pathway. In practice, yield of ATP production $\eta^R_{ATP}$ is not easy to measure as the efficiency, $\eta^R_e$ whereby $\eta^R_e = b \cdot \eta^R_{ATP}$, where $b$ is a constant denoting the amount of biomass formed per unit of ATP. We represent microbial growth as a linear function of the rate of ATP production (*MacLean et al., 2010*; *Pfeiffer et al., 2001*; *Bauchop and Elsden, 1960*) namely $r \cdot J^{ATP}$, where $r$ is some proportionality constant which we here set to 1.

## Sucrose utilization

Both strains can take up sucrose ($S$) and the rate of sucrose pathway is defined by

$$J^S = \frac{V^S_{max} S}{K^S_m + S}$$

where $V^S_{max}$ denotes the maximal rate of the pathway while $K^S_m$ denotes the respective Michaelis-Menten constant. The pathway rate represents the rate at which product is formed, which in this case is the same as the rate at which substrate is consumed. Therefore throughout this article we refer to $V^R_{max}$ as the maximal rate of resource $R$ uptake and $K^R_m$ as the measure of affinity for resource $R$. The efficiency of the pathway utilising sucrose is denoted by $\eta^S$ and for simplicity we assume it to be a constant.

## Invertase production

Invertase producers secrete invertase, an enzyme which catalyses the hydrolysis of sucrose ($S$) with each sucrose molecule being broken down to two molecules of hexose ($H$), namely one molecule of glucose and one molecule of fructose. Hexose is then transported into the cell and for simplicity our model does not differentiate between glucose and fructose molecules. The rate of conversion of sucrose into hexose ($Inv$) is a saturating function of sucrose concentration taking the following from:

$$Inv = inv \frac{S}{k+S}$$

where $inv$ denotes invertase activity and $k$ is a saturation constant. Invertase is costly to produce, and the cost is denoted by $c$, which for simplicity we assume to be a constant (**Gore et al., 2009**).

## Hexose utilisation

Both strains can take up hexose ($H$) and the rate of sucrose pathway is defined by

$$J^H = \frac{V^H_{max} H}{K^H_m + H}$$

where $V^H_{max}$ denotes the maximal rate of the pathway while $K^H_m$ denotes the respective Michaelis-Menten constant. The efficiency of the pathway utilising hexose is denoted by $\eta^H$ and we assume that $J^H \gg J^S$, allowing for hexose to be the preferential carbon source.

## Self-restraint through efficiency of resource utilisation

The efficiency of the hexose pathway is known to depend on the rate of resource uptake, termed rate efficiency trade-off; therefore $\eta^H$ is a decreasing function of $J^H$ and motivated by **Beardmore et al. (2011)** we assume that

$$\eta^H(J^H) = a_1 + \frac{a_2}{1 + \exp(a_3 + a_4 \cdot J^H)},$$

where $a_i, i = 1..4$ are constants.

To predict densities of the producer and non-producer strain, we deploy a reaction-diffusion model enabling the explicit tracking of resource concentrations and population densities in both space and time. In particular let $N_p(x,t)$ and $N_n(x,t)$ denote the density of producers and non-producers, respectively, at time $t$ and spatial location $x$, $x \in [0, l]$ with $l$ denoting a positive constant. Then the model takes the following form

$$\frac{\partial S}{\partial t} = D_S \frac{\partial^2 S}{\partial x^2} - J^S(N_p + N_n) - Inv \cdot N_p \tag{5a}$$

$$\frac{\partial H}{\partial t} = D_H \frac{\partial^2 H}{\partial x^2} - J^H(N_p + N_n) + 2Inv \cdot N_p \tag{5b}$$

$$\frac{\partial N_p}{\partial t} = D_{N_p} \frac{\partial^2 N_p}{\partial x^2} + (1-c)(\eta^H J^H + \eta^S J^S)N_p \tag{5c}$$

$$\frac{\partial N_n}{\partial t} = D_{N_n} \frac{\partial^2 N_n}{\partial x^2} + (\eta^H J^H + \eta^S J^S)N_n \tag{5d}$$

where $\frac{\partial^2}{\partial x^2}$ is one-dimensional diffusion operator while $D_*$ represent diffusion coefficients for sucrose ($S$), hexose ($H$) and cell biomass ($N$). Due to the molecular size we assume that the rate of movement of sucrose is twice as slow as that of hexose, while cells move at an even slower rate.

We impose no-flux boundary conditions in addition to the following initial conditions: $S(x,0) = S_0$, where $S_0$ is a sucrose supply constant, $H(x,0) = 0$ with $N_p(x,0) = N_{p0}(x)$ and $N_n(x,0) = N_{n0}(x)$ representing an initial distribution of producers and non-producers, respectively. The total initial population density is denoted by $N_0$ and an example of an initial spatial distribution of producers and non-producers is shown in **Figure 4—figure**

*supplement 1a*. The model is simulated for different initial frequencies of producers and non-producers until all resources are exhausted and for each case we record the final total population size.

## The model outcomes

Given the complexities inherent to *in planta* infection studies, the parameters in our model were not inferred from empirical data inferred from empirical data obtained through these experiments. Instead, we use previous work (*MacLean et al., 2010*) to identify a parameter range for the model that produces behaviour consistent with the key observation of our *M. oryzae* infection study; namely that a mixture of producers and non-producers maximises the total population size (*Figure 2a* of the main text). These parameters, listed in the table below, were used because the model qualitatively captures the result in *Figure 2a* (as shown in *Figure 4—figure supplement 1b*), and does so for a certain neighbourhood around these points in parameter space.

| | |
|---|---|
| $V_{max}^S$ | 11 **[mmol sucrose/ (g protein x h)]** |
| $V_{max}^H$ | 100 [mmol hexose/ (g protein x h)] |
| $K_m^S$ | 7 [mMol sucrose] |
| $K_m^H$ | 100 [mMol hexose] |
| *inv* | 77 [mmol sugar/ (g protein x h)] |
| *c* | 0.004 |
| $S_0$ | 29.2 [mMol] |
| $N_0$ | $3 \times 10^{-5}$ [g protein/L] |
| $D_S$ | 0.005 [ $l^2$/h] |
| $D_H$ | 0.01 [ $l^2$/h] |
| $D_N$ | 0.0002 [ $l^2$/h] |
| $\eta^S$ | 0.01 [g protein/ mmol sucrose] |
| *k* | $5 \times 10^{-3}$ [mMol sucrose] |
| $a_1$ | 0.0176 |
| $a_2$ | 0.0318 |
| $a_3$ | −2.2649 |
| $a_4$ | 0.205 |

Due to the challenges associated with in vivo parameter estimation, resource utilization parameters values and the shape of the rate-efficiency functions in the above table were chosen to reflect the values obtained empirically for an invertase production system in *S. cereviseae* (*Pfeiffer et al., 2001*).

The cost of invertase production for *M. oryzae* was chosen to be lower than previously estimated for *S. cerevisiae* (*Gore et al., 2009*; *MacLean et al., 2010*). This lower cost experienced can be attributed to the secretome of *M. oryzae* predicted to be much larger than that of *S. cerevisiae* both in terms of total gene products (1885 compared to 424) and as a proportion of total genes (14.51% compared to 6.32%) (*Choi et al., 2010*) meaning that the burden of metabolising a single product would be proportionally less. In addition, invertase production by *S. cerevisiae* is regulated by the experienced glucose environment (*Dodyk and Rothstein, 1964*), whereas for *M. oryzae*, we found it to be upregulated in sucrose environments (*Figure 1c* of the main text). Therefore, *S. cerevisiae* may produce invertase wastefully in a manner that increases its overall cost because it is produced in environments which yield no products.

While the diffusion coefficients have not been obtained empirically we find that the model results are robust to changes in these parameter values as long as they are in a range which allows for sufficient spatial interactions between the producers and non-producers.

The outcome in *Figure 4—figure supplement 1b* could be explained by the fact that when producers are common, the invertase production results in a large spike, both spatial and temporal, in hexose. This enables rapid but inefficient growth. However if a fraction of non-producers is introduced into the population the hexose spike around producers that are in a vicinity of non-producers is smaller, so that the population burns finite resources more efficiently.

Next, to test that the synergy between public goods production and self-restraint drives the result shown in *Figure 4—figure supplement 1b*, we remove the effect of the rate-efficiency trade-off from the model leaving only the public goods dilemma at play. In particular we assume that the efficiency of the hexose pathway ($\eta^H$) is a constant and find that the model predictions revert to the classical finding whereby total population size is maximised for populations containing only producers (*Figure 4—figure supplement 1c*). In this case, the hexose spike created by the producers does not lead to inefficient growth due to the absence of the rate-efficiency trade-off. Note that apart from physically removing the rate-efficiency trade-off from the model, inefficient growth can also be avoided by assuming a low initial sucrose concentration ($S_0$) in the environment.

Moreover, in spatially homogeneous environments both producers and non-producers share resources equally. Therefore the boost of efficiency in resource consumption observed on the boundary between producers and non-producer in spatially structured environments, will not take place in the absence of spatial structure. Indeed, performing simulations of the model (1) in spatially homogeneous environments ($D_S = D_H = D_N = 0$) we again recover the result that the total population size is maximised for a population containing only producers (*Figure 4—figure supplement 1d*).

Since the initial sucrose concentration and spatial structure can be manipulated for in vitro environments, this enabled us to experimentally verify the qualitative predictions of our model (shown in *Figure 4* of the main text).

