## [Decision Letter]

Thank you for submitting your work entitled "Harbouring low virulence strains within a pathogen population can increase both fitness and virulence" for further consideration at *eLife*. Your article has been favorably evaluated by Ian Baldwin (Senior editor), Merijn Kant (Reviewing editor), and three reviewers.

The manuscript by Lindsay and colleagues makes key advances in understanding the role of social interactions in the context of pathogenic infections. The study demonstrates that invertase expression is a cooperative behavior that is subject to the spread of non-producing mutants. In addition, the authors were able to show that low levels of this mutant actually increases both virulence and pathogen population size in the plant host.

While we agreed that the data are intriguing, we did not always agree on the soundness of the data interpretation or on the urgency of additional experiments to support these interpretations. There was no consensus on the question if the evidence for frequency-dependent selection was sufficiently convincing and we would have preferred a more detailed exploration of the frequency dependence across a broader range of frequencies. However, the outcome of the discussion was that the majority of the referees felt that additional experiments will not be necessary. Therefore, we ask for textual changes and some minor changes in your analyses.

1) We feel it is important to check whether the parameter values that emerge from adjusting the model to obtain the behavior in Figure 2 are reasonable. We realize that not all the parameters are known or can be inferred. However, we ask you to add a discussion of whether each parameter value is in the range of what is physically or biologically reasonable, judging for example from values that are known for similar systems.

2) A substantial part of the referee discussion revolved around the question if this invertase can be called a virulence factor or not. This element is clearly confusing and therefore distracting. In literature, some authors discriminate two types of virulence: (I) 'direct virulence factors' that directly interact with the host to get the infection done and (II) 'indirect virulence factors' that influence the wellbeing of the pathogen in the infection site-environment (to cope with the physical conditions, host factors, etc.) and thus also influence pathogen virulence but indirectly. We feel the invertase qualifies as "indirect virulence factor". For example see Figure 6 in Siddiqui & Khan (2012), Parasites & Vectors 5: 6. We ask you to make this explicit in the Introduction around the third paragraph.

3) In relation to point 2 we feel it will be useful to indicate that the public-good scenario you sketch – i.e. increased virulence when the strains are mixed – may not happen to pathogens that use a different infection strategy: the public-good scenario may be realistic for pathogens that deposit their virulence factors (the direct or indirect ones) into the public domain but may not be realistic for pathogens that inject virulence factors directly into individual host cells. Mentioning this in one or two sentences will suffice.

4) We also discussed the following alternative scenario that could explain your data. It could be that the invertase knock-out mutation affects virulence negatively 'simply' because the mutant is less 'healthy/fit' than the wildtype. When mixed with the non-mutated wildtype the latter may compensate for the mutant's defect via the public good. The mutant may now even have more energy available (since it is not making the invertase) to fuel its virulence, than the wildtype. This may increase the overall virulence – in principle independent from food competition – and also in a frequency dependent manner. Since the referees did not agree on the urgency of including this alternative scenario already from the start in your manuscript or only later on, we decided to ask you to address it in the Discussion and there to explain why you feel that the scenario you sketch in the fourth paragraph of the Introduction is more plausible than this alternative.

5) We feel that the current title too much suggests that there is causality between the "low virulence" of the mutant and the increase in the population's virulence when mixed. For accuracy we ask you to replace "low virulence strains" with "public good mutants" (also in the Abstract).

6) We ask you to change 'correctly predicts' in the Abstract with 'suggests';

7) We ask you to replace 'non-producing cheats' in the third and fourth paragraphs of the Introduction with 'non-producing mutants' (because 'cheaters' in the absence of non-cheaters do not cheat);

8) We noticed some quite one-sided t-tests (e.g. legends of Figure 1 and Figure 4) but it is not clear why these are one-sided. Please replace them with two-sided tests or motivate this choice within the manuscript text.

[Editors’ note: a previous version of this study was rejected after peer review, but the authors submitted for reconsideration. The first decision letter after peer review is shown below.]

Thank you for submitting your work entitled "Harbouring low virulence cheats within a pathogen population increases both fitness and virulence" for consideration by *eLife*. Your article has been reviewed by three peer reviewers, and the evaluation has been overseen by Merijn Kant as the Reviewing Editor and Ian Baldwin as the Senior Editor. The following individuals involved in the review of your submission have agreed to reveal their identity: David Queller and Stephen Diggle. A further reviewer remains anonymous. Our decision has been reached after consultation between the reviewers. Based on these discussions and the individual reviews below, we regret to inform you that your work in its present form will not be considered further for publication in *eLife*.

The referees were fascinated by the observation that a mix of invertase and non-invertase producing fungi causes more damage to a plant host than the wild type fungus alone (Figure 2). The referees agreed that this observation is worth perusing and that it could have consequences, for example, for virulence therapy. However, there were substantial concerns regarding the manuscript's framing and the robustness of some of the analyses. Since these concerns may require a substantial amount of work and may affect conclusions we have decided we have to reject your manuscript in its current form. However, we would welcome a revised version in which these concerns have been addressed. I will summarize the main points of discussion below:

1) A large part of the Discussion was dedicated to the question if the invertase is a virulence factor or not – and thus if the mutant really is a virulence cheat – and to which extent this matters for your story. The title reads as if you consider the mutant to be a 'virulence cheat' and later on you refer to the invertase as 'a social virulence factor' but overall you're a bit vague with respect to what virulence exactly means within the context of your work; to which extent the cheater is cheating the virulence process and how this all relates to the observed changes pathogenicity. Clarifying this may come down to merely reframing/rephrasing but this ambiguity was very distracting.

2) The second issue may be a bit more complicated. It was unclear to us why you refer to the mutant as a cheater. Although we agree that the mutant does not contribute to the production a common good, it can only be considered a cheater when this 'non-cooperation' would increase its fitness relative to the non-cheaters. We did not find this back in your data (e.g. in Figure 4 conidia production at 0 and at 1 are equal) and therefore we would like you to justify this term much more explicitly on the basis of the existing or new data.

3) The third issue concerns how you parameterized the cooperation model. It was unclear to us to which extent the parameters are realistic or not and this is important since it determines the strength of the second half of the paper on the rate-yield tradeoff. Hence, the parameterization needs to be justified.

4) Finally, we feel that you are too unclear on several descriptives such as sample sizes, experimental design and on your choice for statistical tests. These things need to be presented clearer. This is especially important because the statistical significance of your key result is borderline. Nevertheless, we are willing to accept the key observation as it is, as long as you (can) explain in the text how everything was randomized properly (otherwise you could get a low p because observations were not all independent). Still we would like you to consider repeating the key experiment to get more confidence on the statistical significance in order to strengthen your claim on its biological significance.

*Reviewer #1:*

With the rise in antibiotic resistance, new strategies to control pathogens are needed. Anti-virulence drugs and strategies such as using 'cheats' to treat infection are two new approaches to tackle the problems we face. Given this, it is always interesting to see a new paper in this area. This new paper suggests we should be cautious in applying anti-virulence techniques because there may be unexpected outcomes as is depicted here.

The finding that mixed infections of cheats and wild-types results in higher virulence is interesting but I have a major issue where I need some convincing. The authors have chosen to work with invertase which liberates carbon sources from plant tissue. I have no issues that invertase can be considered a public good but it's the role in virulence and interpretation of the results that I question.

Invertase mutants grow poorly in a plant and this might be expected as they cannot gain easy access to the carbon source they need to grow. Conversely invertase producers grow well. In mixed cultures, invertase mutants appear to be able to grow and exploit the carbon sources liberated by the producers within the plant. Again, this might be expected if you are showing that invertase is a public good exploitable by cheats.

The interesting finding is then that the virulence of mixed cultures increases. The paper then cautions against using an anti-virulence approach with cheats because here we find an opposite effect. But invertase is not (as far as I know) a virulence factor which damages the host; it helps to promote growth. All the damage to the host will likely be due to specialised virulence factors which presumably the invertase mutant itself can make. So essentially, in a mixed culture, everyone is making virulence factors so the virulence is at least as damaging as a mono-producer infection.

In my opinion, for a true test of whether an anti-virulence strategy works or does not work, you would ideally create a mutant in a known toxin, or a system that directly regulates toxins. Happy to discuss.

*Reviewer #2:*

Invasion of a cooperative infective bacterial population by non-cooperative "cheater" bacteria has been frequently discussed in recent literature as a way to reduce virulence, as an alternative to antibiotic therapy. This manuscript, which uses an experimental system consisting of infections of rice plants by the fungus *M. oryzae*, suggests that the introduction of cheaters can actually enhance virulence in some cases, and suggests a possible explanation for this in terms of an interplay between different cooperative traits.

Specifically, a mutant of *M. oryzae* was generated which fails to produce invertase, but can use invertase produced by wild type cells. Under some conditions introduction of a subpopulation of this mutant was found to increase the virulence of a rice plant infection. To explain this the authors invoke a rate-efficiency tradeoff. They suggest that, in an infection, wild type *M. oryzae* cells produce a large spike of glucose close to them, which they then use inefficiently, producing a relatively small number of spores. However, if cheater cells are present (which do not release glucose) the glucose spike is smaller, meaning that the cells grow more efficiently and produce more spores.

I find the first observation (that introduction of cheaters can increase virulence) interesting, since as the authors note, it contradicts established theory. However, I was not totally convinced by the explanation given for this phenomenon, in terms of the rate-efficiency tradeoff. I feel that to properly demonstrate this as the explanation, more work would need to be done.

Specific points:

1) I would have liked a clearer definition of virulence, and a clearer distinction of how virulence is distinct from fitness. I realize that the experimental measurements of these two quantities are different, but from a theoretical point of view they seem to be assumed to be more or less interchangeable; indeed it is stated in the Introduction that "since public goods aid microbial growth it also affects the extent of damage.…". However virulence factors are specific attributes of a bacterium that cause damage to a host and they can themselves be public goods (e.g. exoenzymes). I felt that a deeper discussion of the distinction between virulence and fitness was needed in the Introduction.

2) Figure 1 does not seem to show what the authors claim it shows. Rather than showing a negative frequency-dependent fitness of the non-producers (cheats) it seems to show the producer fitness. If this figure is indeed as printed then it seems to show that the producer is more fit when at low density, which is completely contradictory to social evolution theory.

3) One important test of whether the mutant is behaving as a cheater is that it should outcompete the wild type in mixed cultures, due to the cost of public goods production. Data is given for this for in vivo infections (Results) but only for one initial cheat frequency. Surely this should also be demonstrated for in vitro growth assays (as in Figure 1) and for a wider range of initial cheat frequencies?

4) The key result is that shown in Figure 2, which shows that the number of spores produced is greater for a mixed infection with 20% cheats than for a wild-type infection (although it is lower for higher cheat frequencies). To explain this the authors show that there is a rate-efficiency tradeoff (Figure 3), which they hypothesize could cause the observed effect due to changes in the size of the glucose spike around the infective population. However, there are multiple reasons why I feel that this explanation is not properly demonstrated:

No comparison is made between the glucose concentrations at which the tradeoff is observed (Figure 3) and the likely concentrations proximal to an infection in the rice plant.

No calculations are made of how much the glucose concentration is likely to be reduced by the presence of the cheaters (this could be done in a simple empirical way using literature values for invertase rates etc.).

In the mathematical model, parameters are chosen to obtain the desired effect (as stated in [Supplementary-material SD2-data]), and no attempt is made to compare these to known values. For example diffusion constants for the sugars: are these realistic? (incidentally the units for these diffusion constants are not properly stated in the table).

The results of the mathematical model (Figure 4—figure supplement 1, panel b) are hard to compare to the experimental results because they are not plotted together and the experimental data (e.g. Figure 2—figure supplement 4) is given on a log scale whereas the model results are linear.

5) I did not follow the rationale of removing spatial heterogeneity by replacing sucrose by glucose (Results, last paragraph): wouldn't this just remove the need for invertase altogether and totally change the relevant social interactions?

*Reviewer #3:*

This paper shows, in a rice-fungus system, how diluting a virulent pathogen with a less virulent one can sometimes lead to greater virulence. It uses experiments and modeling to attribute this effect to the joint action of two different social traits. The result is taken as a caution against indiscriminate use of a dilution strategy to reduce virulence.

The framing of the seems a bit disingenuous, making the result seem more surprising than I think it really was. Although this is a different species than the yeast used in MacLean et al. 2010, it is the same sucrose system and a very similar result (for population fitness, though not virulence). Yet the summary and Introduction read as if this prior study did not exist, instead springing the result and the explanation as a surprise. Still, this is an interesting extension in that it documents the phenomenon for pathogen virulence where it may have some practical relevance.

The main empirical result of the paper is that one of the mixtures (20:80) has higher virulence than the 100% virulent strain treatment (Figure 2). I like the attention given to the fact that this requires a multiple-comparisons Bonferroni correction because there are four relevant comparisons (though I don't think you need all the Boolean algebra in the supplement to make this point). But I do have a little trouble with some aspects of the key p<0.04 result. Where does the n=42 come from? That seems to imply n=7 for each of the 6 treatments (or is that n=21 for each of the two last treatments)? But Figure 2—figure supplement 1 seems to disagree, showing 20 pictures from each of the three pictured treatments. And what were the 3 separate experiments that were pooled to get this figure? A key issue is whether they balanced by treatment – you wouldn't want all of 20-80's in one block for example. That figure also reports a 2-sided t-test in contrast to the 1-sided Mann-Whitney U in Figure 2. Aren't these testing the same comparison and, if so, why are they treated differently?

I would have liked to have seen this key result repeated. It is barely significant. I concede that if p<0.05 is our standard, then I shouldn't demand a repeat every time a result comes close to that. But in this case, if you look at all the data in Figure 2, from 0% Guy11 to 100%, they look oddly behaved. The change in conidia is irregular, first no change, then increase, decrease, increase, and decease. Doesn't that make you a little suspicious that the 20-80 result could be a fluke? You could get that kind of weird fluctuation if not everything was randomized among treatments, such as the spatial position of the plants, so a statement on randomization would be useful.

Figure 2—figure supplement 4 is not a repeat of the same experiment as Figure 2, is it? It seems to show the same overall up-down-up-down pattern, but I can't match up either the conidia values reported in the two figures or the sample sizes. It's very confusing. Why aren't you doing your key Boolean/Bonferroni test of Figure 2 on this set of data, which appears to be larger?

The quadratic curve in Figure 2—figure supplement 4 does not really establish that there is a maximum. Yes, the quadratic fit is demonstrably better than a linear one, but that doesn't mean that this quadratic curve is the real curve. Is it significantly better than a curve that saturates for the two highest values? The real test is the kind that you did earlier, for Figure 2.

---

## [Author Response]

*[…] While we agreed that the data are intriguing, we did not always agree on the soundness of the data interpretation or on the urgency of additional experiments to support these interpretations. There was no consensus on the question if the evidence for frequency-dependent selection was sufficiently convincing and we would have preferred a more detailed exploration of the frequency dependence across a broader range of frequencies. However, the outcome of the discussion was that the majority of the referees felt that additional experiments will not be necessary. Therefore, we ask for textual changes and some minor changes in your analyses.*

*1) We feel it is important to check whether the parameter values that emerge from adjusting the model to obtain the behavior in Figure 2 are reasonable. We realize that not all the parameters are known or can be inferred. However, we ask you to add a discussion of whether each parameter value is in the range of what is physically or biologically reasonable, judging for example from values that are known for similar systems.*

We agree that this may have been unclear and is an important point. A discussion of the parameter values has therefore now been added to Appendix 1, subsection “The model outcomes”, second, third and fourth paragraphs.

*2) A substantial part of the referee discussion revolved around the question if this invertase can be called a virulence factor or not. This element is clearly confusing and therefore distracting. In literature, some authors discriminate two types of virulence: (I) 'direct virulence factors' that directly interact with the host to get the infection done and (II) 'indirect virulence factors' that influence the wellbeing of the pathogen in the infection site-environment (to cope with the physical conditions, host factors, etc.) and thus also influence pathogen virulence but indirectly. We feel the invertase qualifies as "indirect virulence factor". For example see Figure 6 in Siddiqui & Khan (2012), Parasites & Vectors 5: 6. We ask you to make this explicit in the Introduction around the third paragraph.*

Thank you for making us aware of the possible confusion that could be caused by our use of the term virulence factor. We have addressed this point by including clarification of the definitions of ‘direct’ and ‘indirect virulence factors’ in general (Introduction, third paragraph) and also making explicit that in this context invertase represents an indirect virulence factor (Introduction, last paragraph).

*3) In relation to point 2 we feel it will be useful to indicate that the public-good scenario you sketch – i.e. increased virulence when the strains are mixed – may not happen to pathogens that use a different infection strategy: the public-good scenario may be realistic for pathogens that deposit their virulence factors (the direct or indirect ones) into the public domain but may not be realistic for pathogens that inject virulence factors directly into individual host cells. Mentioning this in one or two sentences will suffice.*

To acknowledge this point and clarify the consequences of our finding, we have now explained that multiple trait interactions would not be relevant for virulence factors which act “privately” (Discussion, first paragraph). It is correct, that our observations are concerned with virulence factors that act as public goods.

*4) We also discussed the following alternative scenario that could explain your data. It could be that the invertase knock-out mutation affects virulence negatively 'simply' because the mutant is less 'healthy/fit' than the wildtype. When mixed with the non-mutated wildtype the latter may compensate for the mutant's defect via the public good. The mutant may now even have more energy available (since it is not making the invertase) to fuel its virulence, than the wildtype. This may increase the overall virulence – in principle independent from food competition – and also in a frequency dependent manner. Since the referees did not agree on the urgency of including this alternative scenario already from the start in your manuscript or only later on, we decided to ask you to address it in the Discussion and there to explain why you feel that the scenario you sketch in the fourth paragraph of the Introduction is more plausible than this alternative.*

Thank you for raising the point of a possible alternative mechanism for the synergistic effect that was observed during our study. We have considered this alternative scenario and judge that the evidence obtained to verify predictions made by our mathematical model (Figure 4), does address the possibility of this alternative mechanism. We appreciate that this may have not been explicit enough during the Discussion. Therefore, we have now outlined the alternative hypothesis and explained our reasoning as to why our originally proposed mechanism is indeed the more plausible (Discussion, third paragraph).

*5) We feel that the current title too much suggests that there is causality between the "low virulence" of the mutant and the increase in the population's virulence when mixed. For accuracy we ask you to replace "low virulence strains" with "public good mutants" (also in the Abstract).*

We have made the edits to the title and Abstract as suggested.

*6) We ask you to change 'correctly predicts' in the Abstract with 'suggests';*

This has now been changed as requested.

*7) We ask you to replace 'non-producing cheats' in the third and fourth paragraphs of the Introduction with 'non-producing mutants' (because 'cheaters' in the absence of non-cheaters do not cheat);*

These edits have been incorporated as suggested.

*8) We noticed some quite one-sided t-tests (e.g. legends of Figure 1 and Figure 4) but it is not clear why these are one-sided. Please replace them with two-sided tests or motivate this choice within the manuscript text.*

We have replaced these one-sided tests with two-sided tests.

[Editors’ note: the author responses to the first round of peer review follow.]

*[…] Since these concerns may require a substantial amount of work and may affect conclusions we have decided we have to reject your manuscript in its current form. However, we would welcome a revised version in which these concerns have been addressed. I will summarize the main points of discussion below:*

*1) A large part of the Discussion was dedicated to the question if the invertase is a virulence factor or not – and thus if the mutant really is a virulence cheat – and to which extent this matters for your story. The title reads as if you consider the mutant to be a 'virulence cheat' and later on you refer to the invertase as 'a social virulence factor' but overall you're a bit vague with respect to what virulence exactly means within the context of your work; to which extent the cheater is cheating the virulence process and how this all relates to the observed changes pathogenicity. Clarifying this may come down to merely reframing/rephrasing but this ambiguity was very distracting.*

We apologise for the confusion. This has now been corrected throughout the manuscript and made consistent, as suggested by the editor. Our study demonstrates that invertase production in *Magnaporthe oryzae* conforms to the definition of a cooperative trait (details presented in the first three paragraphs of the Results section). In addition, we also demonstrate that invertase non-producers have lower virulence than invertase producers (Figure 2).

Therefore, for clarity, previously used terminology "low virulence cheats" and "high virulence cooperators" has now consistently been replaced by "low virulence public goods non-producers" and "high virulence public goods producers", respectively. Furthermore, we now consistently refer to "cooperators" and "cheats" only in the context of invertase production.

We have also changed the title to: "Harbouring low virulence strains within a pathogen population can increase both fitness and virulence",and altered the Abstract accordingly, which we hope is now clearer and which we believe will avoid the confusion.

Moreover, we now clearly explain that the pathogen virulence is measured by the area of disease lesion coverage of an infected leaf. The lesions are symptoms of rice blast disease and are a direct sign of damage inflicted upon the host, affecting the growth and yield of the plant (Results, third paragraph).

Finally, our findings and the terminology regarding the invertase production in *M. oryzae*, are consistent with other cooperative systems involving extracellular factors, such as siderophore production (Harrison et al., BMC Biology 2006), quorum sensing (Rumbaugh et al., Current Biology 2009) and toxin production (Raymond et al., Science 2012), where cooperators have been observed to have higher virulence than cheats.

*2) The second issue may be a bit more complicated. It was unclear to us why you refer to the mutant as a cheater. Although we agree that the mutant does not contribute to the production a common good, it can only be considered a cheater when this 'non-cooperation' would increase its fitness relative to the non-cheaters. We did not find this back in your data (e.g. in Figure 4 conidia production at 0 and at 1 are equal) and therefore we would like you to justify this term much more explicitly on the basis of the existing or new data.*

As clarified above and throughout the manuscript, the invertase production in *M. oryzae* conforms to the definition of a cooperative trait (details presented in the first three paragraphs of the Results section) and therefore the invertase producer is termed a cooperator while the invertase non-producer is termed a cheat (Results, fourth paragraph).

Moreover, as requested by the Editor we now include an additional data set showing that, consistent with cooperation theory, the invertase non-producer (the *cheat)* increases its fitness relative to the invertase producer (the *cooperator)* in environments where cheats are found in the vicinity of cooperators (new Figure 1). This is in addition to the finding that the non-producer has a selective advantage over the producer in mixed plant infections (Results, third paragraph).

The old Figure 1 (now moved to Figure 1—figure supplement 3) showed that in sufficiently structured environments, producers and non-producers can coexist, which has also been observed in other invertase cooperation systems (MacLean et al., PloS Biology 2010).

To be completely clear, Figure 4 shows population fitness in glucose (not sucrose). Invertase production in *M. oryzae*, however, is sucrose-induced (Figure 1). Therefore, the concern raised that "conidia production at 0 and 1 frequency of producers is equal" is for growth in glucose, where both strains grow equally well. As such, this data is not relevant when ascertaining the fitness advantage of non-producers over producers, which can only be determined for *M. oryzae* based on growth in sucrose. This has now been clarified (Results, eighth paragraph).

*3) The third issue concerns how you parameterized the cooperation model. It was unclear to us to which extent the parameters are realistic or not and this is important since it determines the strength of the second half of the paper on the rate-yield tradeoff. Hence, the parameterization needs to be justified.*

The reason for developing a mathematical model was so that we would have a tool for generating testable predictions regarding the mechanisms driving our in plantainfection results in Figure 2.

The model needed to capture key features of our experimental system, but it also needed to be sufficiently simple so that we could manipulate and probe the spatio-temporal dynamics of extracellular enzymes, resource concentrations and pathogen population densities. This is vital for generating testable predictions (as discussed in [Supplementary-material SD2-data]).

Given the complexities inherent to in plantainfection studies, the parameters in our model were not inferred from empirical data obtained through these experiments. Instead, we used previous work (MacLean et al., PloS Biology 2010) to identify a parameter range for the model that produced behaviour consistent with the key observation of the *M. oryzae* system shown in Figure 2. Those parameters were used because the model qualitatively captured the result in Figure 2, and did so for a certain neighbourhood around this point in parameter space. Given this (Figure 4—figure supplement 1), we then explored parameter changes that could destroy this result, meaning moving outwards towards the boundary of that neighbourhood in parameter space.

From doing this, we predicted that the interaction between resource availability aided by invertase production and the rate-efficiency trade-off is a driver of the result in Figure 4—figure supplement 1. We say this because this result can be destroyed in the model simulations by (1) weakening the impact of the trade-off on pathogen growth (Figure 4—figure supplement 1) or by (2) altering the spatial structure of the environment (Figure 4—figure supplement 1).

Since (1) and (2) can be manipulated in vitro, we subsequently conducted experiments with *M. oryzae* (Figure 4) and the outcome was consistent with the model predictions. This experimental verification is, we claim, what determines "the strength of the second half of the paper" as it provides empirical evidence of the role of multiple social interactions between pathogens in the success of a virulence reduction strategy.

A more detailed description of the modelling procedure and its main purpose is now included in [Supplementary-material SD2-data].

*4) Finally, we feel that you are too unclear on several descriptives such as sample sizes, experimental design and on your choice for statistical tests. These things need to be presented clearer. This is especially important because the statistical significance of your key result is borderline. Nevertheless, we are willing to accept the key observation as it is, as long as you (can) explain in the text how everything was randomized properly (otherwise you could get a low p because observations were not all independent). Still we would like you to consider repeating the key experiment to get more confidence on the statistical significance in order to strengthen your claim on its biological significance.*

We have now addressed all of these points. Sample sizes are now clarified in Materials and methods section (subsection “Pathogenicity and in planta fitness assay of M. oryzae”, second and third paragraphs). We use a standard notation "n" to denote the number of replicates per each condition tested. In particular, in plantainfection experiment determining the number of conidia per lesion (Figure 2) has n=42 replicates per each of the 6 conditions assessed, totalling 252 data points (i.e. disease lesions assessed). On the other hand, a different in plantainfection experiment determining the size of disease lesions (Figure 2) has n=20 replicates per each of the 3 conditions assessed, totalling 60 data points (i.e. disease lesions assessed).

The choice of statistical tests for pairwise comparisons was made based on whether the data satisfies a normal distribution and/or equal variance tests (detailed in subsection “Data Analysis”, second paragraph). For normally distributed data a 2-sample t-test (for either equal or unequal variances) was used, otherwise a Mann-Whitney U test was deployed.

Moreover, we also make a further clarification regarding the choice of statistical tests for multiple comparisons (detailed in subsection “Data Analysis”, third paragraph). If the data violated the assumptions of ANOVA, the non-parametric Kruskal Wallis test was performed, followed by two-sided Mann Whitney U test with Bonferroni corrections. Otherwise one-way ANOVA was used followed by two-sided t-test with Bonferonni corrections.

Our experimental design is now described in full in a new Supplementary file 3, demonstrating that the observations are independent of each other. We also stress the statistical significance of our two key results:

Result 1: Infecting populations composed of the highly virulent strain alone were not the fittest. This can be concluded from the in vivoinfection data presented in Figure 2 (p<0.0365, two-sided Mann- Whitney U test with Bonferroni correction, n=42). Note that due to the nature of the data in Figure 2, a non-parametric test was performed. To improve normality of the data and satisfy homogeneity of variance we also log-transform the data in Figure 2 as per previous studies involving microbial pathogens (Lu and Collins, PNAS 2007; Köhler et al., PloS Pathogens 2009, Raymond et al., Science 2012). Applying a parametric test to the log-transformed data increases the significance of Result 1 as detailed in Figure 2—figure supplement 4 (p<0.017, two-sided t-test with Bonferroni correction, n=42). We now report both analyses (subsection “Data Analysis”, fourth paragraph and Figure 2—figure supplement 4).

*Even more importantly for disease management*, our study also shows that:

Result 2: *Infecting populations composed of the highly virulent strain alone were not the most virulent.* This can be concluded from the in vivoinfection data presented in Figure 2 (p<0.0032, two-sided t-test with unequal variances, n=20). Note that in this case the data in Figure 2 meets parametric test assumptions.